# Endoplasmic reticulum-associated SKN-1A/Nrf1 mediates a cytoplasmic unfolded protein response and promotes longevity

Nicolas J Lehrbach[1,2], Gary Ruvkun[1,2]*

[1]Department of Molecular Biology, Massachusetts General Hospital, Boston, United States; [2]Department of Genetics, Harvard Medical School, Boston, United States

**Abstract** Unfolded protein responses (UPRs) safeguard cellular function during proteotoxic stress and aging. In a previous paper (Lehrbach and Ruvkun, 2016) we showed that the ER-associated SKN-1A/Nrf1 transcription factor activates proteasome subunit expression in response to proteasome dysfunction, but it was not established whether SKN-1A/Nrf1 adjusts proteasome capacity in response to other proteotoxic insults. Here, we reveal that misfolded endogenous proteins and the human amyloid beta peptide trigger activation of proteasome subunit expression by SKN-1A/Nrf1. SKN-1A activation is protective against age-dependent defects caused by accumulation of misfolded and aggregation-prone proteins. In a *C. elegans* Alzheimer's disease model, SKN-1A/Nrf1 slows accumulation of the amyloid beta peptide and delays adult-onset cellular dysfunction. Our results indicate that SKN-1A surveys cellular protein folding and adjusts proteasome capacity to meet the demands of protein quality control pathways, revealing a new arm of the cytosolic UPR. This regulatory axis is critical for healthy aging and may be a target for therapeutic modulation of human aging and age-related disease.

DOI: https://doi.org/10.7554/eLife.44425.001

**\*For correspondence:**
ruvkun@molbio.mgh.harvard.edu

**Competing interests:** The authors declare that no competing interests exist.

## Introduction

Loss of proteostasis and accumulation of damaged and misfolded proteins is a hallmark of aging (*López-Otín et al., 2013*). Cells detect protein misfolding and activate unfolded protein responses (UPRs) that adjust protein metabolism to restore proteostasis (*Pilla et al., 2017*). These changes include inhibition of translation to limit synthesis of new proteins, upregulation of chaperones that mediate protein folding, and enhanced destruction of misfolded proteins via the proteasome or autophagy. Protein damage that accrues over time appears to eventually overcome these homeostatic mechanisms and contributes to the decline in cellular and organismal health during aging. Mutations that persistently increase production of unfolded proteins or that impair their clearance accelerate this process to cause a number of adult-onset neurodegenerative diseases (*Hipp et al., 2014*). Conversely, increasing the activity of UPR pathways to enhance proteostasis may be a means to combat these diseases or even aging itself (*Powers et al., 2009*; *Taylor and Dillin, 2011*).

The proteasome mediates the targeted degradation of misfolded and damaged proteins and is essential for proteostasis and cell viability (*Collins and Goldberg, 2017*). Impaired proteasome function is associated with aging and age-dependent neurodegenerative diseases (*Saez and Vilchez, 2014*). The SKN-1A/Nrf1 transcription factor regulates the transcription of proteasome subunit genes to increase proteasome biogenesis when the proteasome is inhibited, for example by proteasome inhibitor drugs (*Grimberg et al., 2011*; *Lehrbach and Ruvkun, 2016*; *Radhakrishnan et al., 2010*; *Steffen et al., 2010*). This compensatory response is essential for the survival of mammalian cells and *C. elegans* under conditions of impaired proteasome function (*Lehrbach and Ruvkun, 2016*; *Radhakrishnan et al., 2010*). SKN-1A/Nrf1 is an unusual transcription factor that associates

with the ER via an N-terminal transmembrane domain (*Glover-Cutter et al., 2013*; *Wang and Chan, 2006*). The bulk of SKN-1A/Nrf1 extends into the ER lumen where it undergoes N-linked glycosylation at particular asparagine residues (*Radhakrishnan et al., 2014*; *Zhang et al., 2007*). After it is glycosylated, SKN-1A/Nrf1 is translocated from the ER lumen to the cytoplasm by the ER-associated degradation (ERAD) machinery, which also targets this short half-life transcription factor for rapid proteasomal degradation (*Lehrbach and Ruvkun, 2016*; *Steffen et al., 2010*). Under conditions of impaired proteasome function, the SKN-1A/Nrf1 half-life is dramatically increased so that some of the protein escapes degradation and enters the nucleus where it can up-regulate target genes (*Lehrbach and Ruvkun, 2016*; *Li et al., 2011*; *Radhakrishnan et al., 2010*; *Steffen et al., 2010*). All proteasome subunit genes are direct transcriptional targets of SKN-1A/Nrf1 (*Niu et al., 2011*; *Sha and Goldberg, 2014*). Activation of SKN-1A/Nrf1 also requires deglycosylation by the peptide N-glycanase PNG-1/NGLY1 and proteolytic cleavage by the aspartic protease DDI-1/DDI2 (*Koizumi et al., 2016*; *Lehrbach and Ruvkun, 2016*; *Tomlin et al., 2017*). It is not yet known whether the SKN-1A/Nrf1 transcription factor regulates proteasome levels in response to other proteotoxic insults.

Here we show that SKN-1A increases proteasome subunit gene expression in response to endogenous misfolded proteins or expression of a foreign aggregation-prone protein, the human amyloid beta peptide. This pathway requires the DDI-1/DDI2 aspartic protease and the PNG-1/NGLY1 peptide N-glycanase, factors that are also required for activation of SKN-1A during proteasome dysfunction. *C. elegans* mutants that lack SKN-1A show enhanced age-dependent toxicity of misfolding proteins, accelerated tissue degeneration during aging and reduced overall lifespan. Conversely, increasing SKN-1A levels is sufficient to extend *C. elegans* lifespan. Our data suggests that SKN-1A/Nrf1 mediates an unfolded protein response that adjusts proteasome capacity to ensure protein quality control. This pathway preserves cellular function during aging by limiting accumulation of unfolded and damaged proteins.

## Results

### Misfolded proteins trigger SKN-1A activation

A transgene expressing GFP from the *rpt-3* proteasome subunit gene promoter shows SKN-1A-dependent upregulation in response to proteasome dysfunction (*Lehrbach and Ruvkun, 2016*). To explore the genetic defects that can activate such proteasome response pathways and the mechanisms that control proteasome subunit gene expression, we performed a large-scale random EMS-mutagenesis screen for mutants that cause increased expression of *rpt-3p::gfp*. We isolated 21 alleles affecting proteasome subunit genes, including mutations affecting components of the 19S regulatory particle and the 20S catalytic core of the proteasome (*Table 1*, *Figure 1—figure supplement 1*). Many of the mutants show temperature sensitive defects in fertility, consistent with previous genetic analysis of proteasome function in *C. elegans* germline development (*Figure 1—figure supplement 2*) (*Shimada et al., 2006*). Some proteasome mutant strains show severe temperature sensitive developmental defects that may reflect temperature-sensitivity of the mutant protein (*Table 1*). Activation of *rpt-3p::gfp* in proteasome hypomorphic mutants requires *skn-1* and depletion of SKN-1 by RNAi causes larval lethality in all but one of the mutant strains, although *skn-1 (RNAi)* is not larval lethal in wild type (*Table 1*, *Figure 1—figure supplement 1*). These data indicate that a wide range of perturbations to proteasome function trigger SKN-1A activation and confirm that compensatory upregulation of proteasome subunit genes by SKN-1A is critical for survival of proteasome dysfunction, either due to mutations or pharmacological inhibition (*Keith et al., 2016*; *Lehrbach and Ruvkun, 2016*).

Our large genetic screen also identified three alleles of *unc-54*, which encodes a myosin class II heavy chain (MHC B) expressed in the body wall muscle (*Ardizzi and Epstein, 1987*; *Epstein et al., 1974*). UNC-54 is the major MHC B in body wall muscles and *unc-54* loss of function mutations cause paralysis. The *unc-54* alleles we isolated activate *rpt-3p::gfp* specifically in body wall muscle cells (*Figure 1a,b*), unlike the proteasome mutations which activate *rpt-3p::gfp* in many tissues. To understand how MHC B affects *rpt-3p::gfp*, we tested other *unc-54* alleles. The temperature-sensitive *unc-*

**Table 1.** Protesome subunit mutants.

| genotype | allele effect | Viability at 20°C | Viability at 25°C | rpt-3p::gfp induction on skn-1(RNAi) | growth on skn-1(RNAi) |
|---|---|---|---|---|---|
| wild type | + | Yes | Yes | | + |
| pas-1(mg511) | G82R | Yes | No (Ste) | lost | Lva |
| pbs-2(mg581) | C90Y | Yes | Yes | lost | Lva |
| pbs-2(mg538) | G93E | Yes | Yes | lost | Lva |
| pbs-2(mg530) | D97N | Yes | ND | lost | Lva |
| pbs-3(mg527) | S180L | Yes | Yes | lost | Lva |
| pbs-4(mg539) | M48K | Yes | No (Emb/Lva) | lost | Lva |
| pbs-5(mg509) | 3'UTR | Yes | Yes | lost | Lva |
| pbs-5(mg502) | promoter* | Yes | Yes | lost* | +* |
| rpt-6(mg513) | I302N, P328S | Yes | Yes | lost | Lva |
| rpt-6(mg512) | E278K | Yes | No (Ste) | lost | Lva |
| rpn-1(mg514) | S519F | Yes | No (Ste) | lost | Lva |
| rpn-1(mg537) | G431E | Yes | No (Ste) | lost | Lva |
| rpn-5(mg534) | T76I | Yes | No (Emb/Lva) | lost | Lva |
| rpn-8(mg587) | G73R | Yes | No (Ste) | lost | Lva |
| rpn-8(mg536) | A88V | Yes | No (Ste) | lost | Lva |
| rpn-9(mg533) | G357STOP | Yes | No (Emb/Lva) | lost | Lva |
| rpn-10(mg525) | G114E | Yes | No (Ste) | lost | Lva |
| rpn-10(mg495) | K130STOP | Yes | No (Ste) | lost | Lva |
| rpn-10(mg531) | Frameshift | Yes | ND | lost | Lva |
| rpn-10(mg529) | Q298STOP | Yes | ND | lost | Lva |
| rpn-11(mg494) | E108K | Yes | No (Ste) | lost | Lva |

ND: Not determined

* *Lehrbach and Ruvkun, 2016*

DOI: https://doi.org/10.7554/eLife.44425.002

54(e1301) and unc-54(e1157) alleles encode mutant forms of UNC-54/MHC B that are prone to misfold and aggregate (*Ben-Zvi et al., 2009*; *Gidalevitz et al., 2006*; *Silva et al., 2011*). Both unc-54 (e1301) and unc-54(e1157) activate expression of rpt-3p::gfp in muscle cells (*Figure 1a,b*). By contrast, unc-54(e190), a null (deletion) allele that eliminates MHC B expression and causes paralysis regardless of temperature (*Dibb et al., 1985*), does not activate rpt-3p::gfp (*Figure 1a,c,e*). Interestingly, all of the unc-54 alleles we isolated in our screen for proteasome subunit activation are missense mutations that cause temperature-sensitive paralysis similarly to unc-54(e1301) (*Figure 1c*, *Figure 1—figure supplement 3*). These data suggest that rpt-3p::gfp activation is triggered by the presence of mutant forms of MHC B that are prone to misfold, not simply by loss of MHC B or defective muscle function.

Activation of rpt-3p::gfp expression by temperature-sensitive mutant MHC B is completely lost in skn-1a(mg570) mutant animals that lack SKN-1A but retain other SKN-1 isoforms (*Figure 1d,e*). To test for activation of SKN-1A at the protein level, we used a transgene to ubiquitously express a truncated form of SKN-1A that lacks the DNA binding domain and is fused to GFP at the C-terminus (rpl-28p::skn-1a[ΔDBD]::gfp). This protein undergoes the same post-translational regulation as full length SKN-1A (*Lehrbach and Ruvkun, 2016*). We found increased levels of SKN-1A[ΔDBD]::GFP accumulates specifically in the body wall muscle cells of unc-54(e1301) and unc-54(mg519) animals but not in the wild type (*Figure 1—figure supplement 4*). We conclude that expression of temperature-sensitive mutant UNC-54/MHC B triggers rpt-3p::gfp expression via activation of SKN-1A.

Activation of proteasome subunit expression in animals expressing mutant MHC B might reflect a general response to accumulation of misfolded proteins. To test the model that unfolded proteins

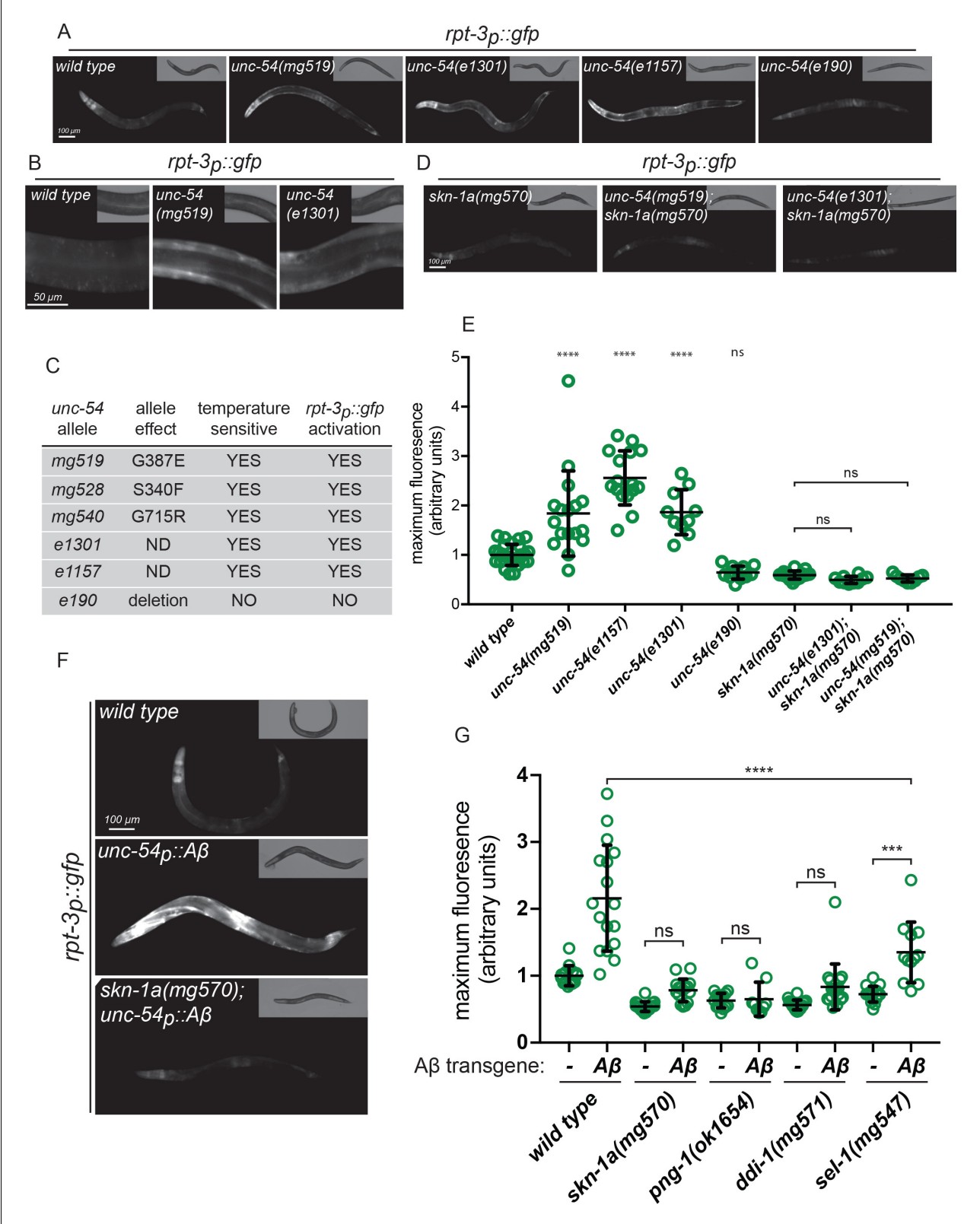

**Figure 1.** Misfolded proteins activate SKN-1A. (**a, b**) Fluorescence images showing *rpt-3p::gfp* expression in various *unc-54* mutants. (**c**) Temperature dependent paralysis and *rpt-3p::gfp* effects of *unc-54* alleles. (**d**) Fluorescence images showing *rpt-3p::gfp* induction in *unc-54(mg519)* and *unc-54 (e1301)* requires *skn-1a*. (**e**) Quantification of *rpt-3p::gfp* expression in various *unc-54* mutants. (**f**) Fluorescence images showing Aβ expression in muscle increases *rpt-3p::gfp* fluorescence in wild type but not in *skn-1a* mutant animals. (**g**) Quantification of Aβ-induced activation of *rpt-3p::gfp* in various

*Figure 1 continued on next page*

*Figure 1 continued*

mutant backgrounds. Panels e and g: ****p<0.0001; ***p<0.001; ns p>0.05. (one-way ANOVA with Tukey's multiple comparison test), P-value compared to wild type unless otherwise indicated.

DOI: https://doi.org/10.7554/eLife.44425.003

The following figure supplements are available for figure 1:

**Figure supplement 1.** Proteasome subunit mutations that activate SKN-1A.
DOI: https://doi.org/10.7554/eLife.44425.004
**Figure supplement 2.** Fertility defects of proteasome subunit mutant strains.
DOI: https://doi.org/10.7554/eLife.44425.005
**Figure supplement 3.** Temperature sensitive paralysis of *unc-54* mutants.
DOI: https://doi.org/10.7554/eLife.44425.006
**Figure supplement 4.** *unc-54ts* mutants activate SKN-1A.
DOI: https://doi.org/10.7554/eLife.44425.007
**Figure supplement 5.** Increased expression of *rpt-3p::gfp* in *hsf-1* mutants.
DOI: https://doi.org/10.7554/eLife.44425.008

engage SKN-1A, we examined the response to another misfolded protein, the human amyloid beta peptide (Aβ). Aβ is derived from the posttranslational processing of the Amyloid precursor protein (APP). Mutations that increase production of Aβ or impair its clearance are associated with Alzheimer's disease. In Alzheimer's disease, Aβ forms aggregates that may play an important role in pathogenesis (*Selkoe and Hardy, 2016*). Transgenic *C. elegans* that express human Aβ in muscle cells (*unc-54p::Aβ*) show adult-onset defects in muscle function and serve as a model for the cell biology of Aβ accumulation and toxicity (*Link, 2006*). We found that Aβ expression in muscle triggers strong muscle-specific activation of *rpt-3p::gfp*, which is lost in *skn-1a(mg570)* mutant animals that lack the transmembrane-domain-containing Nrf1 orthologue SKN-1A (*Figure 1f*).

To test whether SKN-1A activation is broadly associated with protein folding defects, we monitored *rpt-3p::gfp* activation in *hsf-1(sy441)* heat shock transcription factor mutants. *hsf-1* encodes the *C. elegans* orthologue of HSF1, which regulates expression of multiple cytoplasmic chaperones under proteotoxic stress conditions such as elevated temperature (*Fujimoto and Nakai, 2010*). *hsf-1(sy441)* is a hypomorphic allele that disrupts chaperone regulation (*Hajdu-Cronin et al., 2004*). *hsf-1(sy441)* mutant animals develop normally at lower temperatures, but arrest larval development at 25°C, presumably due to the toxic accumulation of misfolded proteins in the cytoplasm. *hsf-1(sy441)* L4 larvae raised at 20°C show unaltered expression of *rpt-3p::gfp* compared to the wild type. However, *rpt-3p::gfp* expression is significantly increased in *hsf-1* mutant animals following upshift to 25°C for 24 hr (*Figure 1—figure supplement 5*). This activation of *rpt-3p::gfp* in the *hsf-1* mutant requires SKN-1A (*Figure 1—figure supplement 5*). These results indicate that SKN-1A is broadly activated under conditions that increase the cellular burden of unfolded proteins. It is therefore likely that there are many endogenous proteins that, when misfolded, are able to trigger a SKN-1A-dependent response. The effects of the *unc-54ts* mutants and *unc-54p::Aβ* indicate that this response is sensitive enough to detect a single - albeit abundant - unfolded protein. Further, at least in muscle, the response is cell autonomously elicited by protein misfolding, but not by mutations - such as the *unc-54(e190)* deletion - that severely compromise muscle function without misfolded protein expression. This proteasomal response therefore does not depend on cellular or organismal consequences of tissue dysfunction in general. Taken together these data strongly suggest that SKN-1A is activated as part of a cell-autonomous response to cytoplasmic unfolded proteins.

The peptide:N-glycanase PNG-1/NGLY1, the aspartic protease DDI-1/DDI2 and the ERAD component SEL-1/SEL1 are each necessary to activate SKN-1A in response to direct proteasomal insults (*Lehrbach and Ruvkun, 2016*). To determine if this same genetic pathway is necessary to activate SKN-1A in response to misfolded proteins, we measured activation of *rpt-3p::gfp* by Aβ in *png-1*, *ddi-1*, and *sel-1* mutants. The SKN-1A-dependent *rpt-3p::gfp* transcriptional response to Aβ is lost in *png-1(ok1654)* and *ddi-1(mg571)* mutants and is diminished in *sel-1(mg457)* mutants (*Figure 1g*). We conclude that related, or possibly identical, mechanisms govern SKN-1A activation by both direct assaults on the proteasome and the presence of misfolded and/or aggregated proteins.

# SKN-1A is cell autonomously activated by impaired proteasome function

These data suggest that SKN-1A mediates a cell-autonomous transcriptional response to protein misfolding in muscle cells. SKN-1A also responds to proteasome dysfunction, but whether this response is cell autonomous is not known. We therefore configured a system to induce cell-type specific impairment of proteasome function in body wall muscle cells. Over-expression of an active site mutant of the β5 subunit of the 20S proteasome in otherwise wild-type cells causes proteasome dysfunction in yeast and the mouse (*Heinemeyer et al., 1997*; *Li et al., 2004*). We generated a transgene that expresses the corresponding active site mutant of the *C. elegans* β5 subunit, PBS-5[T65A], under control of the muscle specific *myo-3* promoter (*myo-3p::pbs-5*[*T65A*]), such that proteasome dysfunction is induced specifically in muscle cells.

The *myo-3p::pbs-5*[*T65A*] transgene causes muscle-specific activation of the *rpt-3p::gfp* proteasome subunit reporter in a manner closely resembling that caused by mutant MHC B and Aβ (*Figure 2a,b*). This activation is lost in *skn-1a*(*mg570*) mutants, consistent with a SKN-1A-dependent response to proteasome dysfunction (*Figure 2a*). Wild type animals carrying the *myo-3p::pbs-5*[*T65A*] transgene show mildly impaired locomotion compared to non-transgenic controls (*Figure 2c*). Because impairment of the proteasome may cause age-dependent defects in cellular function, we examined movement of these animals at different ages. The locomotor rate of wild type animals carrying the *myo-3p::pbs-5*[*T65A*] transgene is reduced to a similar extent in day 1 and day 7 adults showing that this mild defect is not exacerbated by age (*Figure 2c*). This suggests that

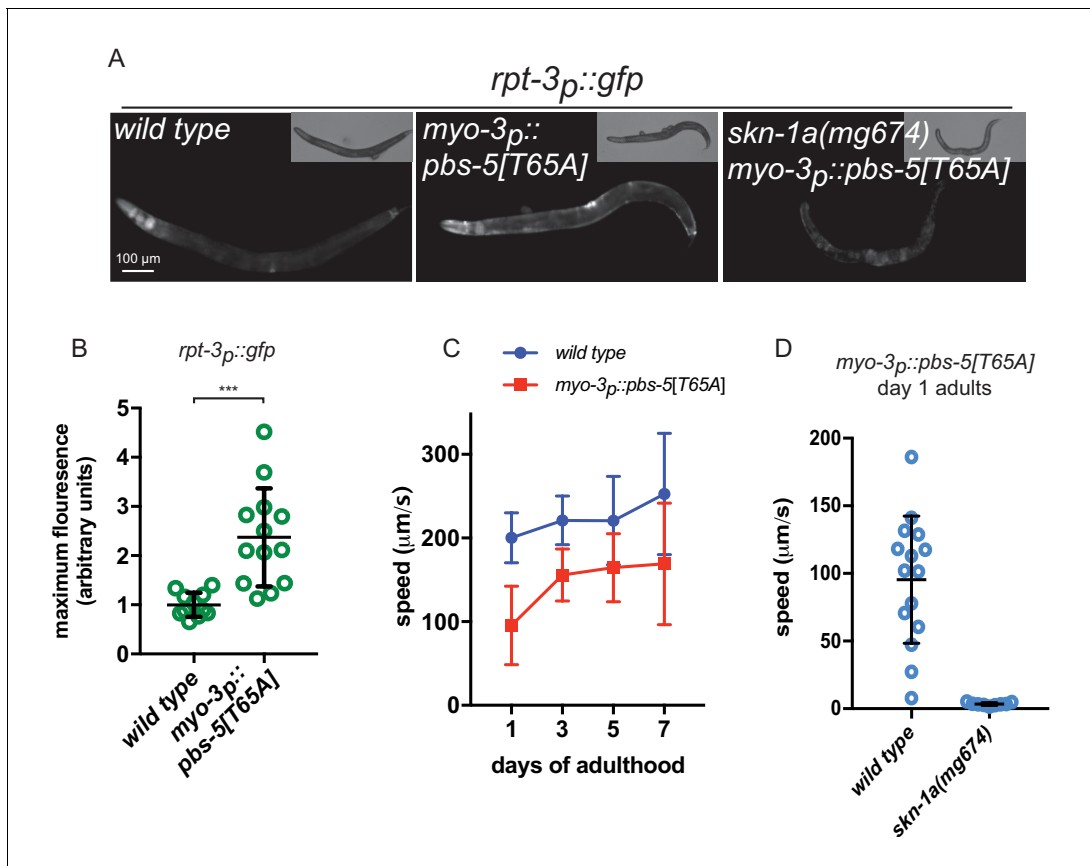

**Figure 2.** Proteasome impairment in muscle causes cell autonomous activation of SKN-1A. (a) Fluorescence images showing *rpt-3p::gfp* expression in animals expressing a dominant negative proteasome subunit in the muscle (*myo-3p::pbs-5*[*T65A*]). (b) Quantification of *rpt-3p::gfp* expression in animals expressing a mutant proteasome subunit in the muscle. ***p<0.001 (Welch's t-test). (c) Comparison of locomotor rate between wild type and *myo-3p::pbs-5*[*T65A*] transgenic animals. (d) Comparison of locomotor rate between wild type and *skn-1a* mutant animals carrying the *myo-3p::pbs-5*[*T65A*] transgene on day 1 of adulthood.
DOI: https://doi.org/10.7554/eLife.44425.009

wild-type muscle cells are robust to proteasomal insults and so are able to maintain near-normal function despite the presence of the mutant β5 subunit. By contrast, the *myo-3p::pbs-5[T65A]* transgene causes complete paralysis in *skn-1a(mg570)* mutant animals lacking the SKN-1A-mediated proteasomal response pathway (*Figure 2d*). We conclude that SKN-1A mediates cell-autonomous activation of proteasome subunit genes in response to proteasome impairment, and that this SKN-1A-dependent compensation is essential for maintaining function in muscle cells experiencing proteasome dysfunction.

## SKN-1A activation by misfolded proteins involves little or no impairment of proteasome function

Aggregation-prone proteins including human Aβ may interact with proteasomes and impair their function (*Ayyadevara et al., 2015*; *Deriziotis et al., 2011*; *Gregori et al., 1995*; *Kristiansen et al., 2007*; *Snyder et al., 2003*). To test the possibility that misfolded proteins trigger SKN-1A activation via inhibition of the proteasome, we generated a reporter of proteasome activity, a ubiquitously expressed unstable ubiquitin-GFP fusion protein (*rpl-28p::ub(G76V)::gfp*). The UB(G76V)::GFP ubiquitin fusion protein is normally degraded by the proteasome, but accumulates to detectable levels if proteasome function is impaired (*Johnson et al., 1995*; *Segref et al., 2014*). As expected, this reporter of proteasome activity reveals a muscle-specific proteasomal defect in *myo-3p::pbs-5[T65A]* transgenic animals (*Figure 3a,b*). Thus tissue-specific impairment of the proteasome in body wall muscle can be readily detected by monitoring UB(G76V)::GFP levels. Stabilization of UB(G76V)::GFP in PBS-5[T65A]-expressing muscle cells is greatly enhanced in the *skn-1a* mutant – all mutant animals show accumulation of GFP in all muscle cells and at higher levels than the wild type (*Figure 3a,b*). These data show that the SKN-1A transcriptional program partially corrects the muscle proteasomal defect caused by the *myo-3p::pbs-5[T65A]* insult. The severe locomotor defects and paralysis of *myo-3p::pbs-5[T65A]* animals that lack SKN-1A therefore likely stem from enhanced defects in proteasome function.

Mutant UNC-54, Aβ and PBS-5[T65A] all cause SKN-1A activation, as indicated by activation of *rpt-3p::gfp*. If all three trigger SKN-1A by the same mechanism – that is, by impairing proteasome function – they should also stabilize UB(G76V)::GFP. However, in contrast to *myo-3p::pbs-5[T65A]*, we did not observe stabilization of UB(G76V)::GFP in *unc-54p::Aβ* transgenics (*Figure 3a,b*). Because activation of SKN-1A could compensate for an effect of Aβ on proteasome function, we also examined the effect of Aβ in *skn-1a(mg570)* mutants. *unc-54p::Aβ* only weakly affected UB(G76V)::GFP levels within the muscle cells of *skn-1a* mutants: about 10% of *skn-1a(mg570)* Aβ-expressing animals showed weak accumulation of UB(G76V)::GFP in some muscle cells suggesting a mild impairment of proteasome function (*Figure 3a,b*). We also tested the effect of *unc-54(e1301)* and *unc-54(mg519)* in the *skn-1a(mg570)* mutant background and found no effect on UB(G76V)::GFP degradation in the muscle (*Figure 3b*).

In mammalian cells, UbG76V::GFP accumulates only in cells with severely compromised proteasome function, as measured by Suc-LLVY-AMC hydrolysis in cell lysates (*Dantuma et al., 2000*). It is therefore possible that mutant MHC B and Aβ cause mild defects in proteasome function that are sufficient to activate *rpt-3p::gfp* without altering steady state levels of UB(G76V)::GFP. To test this possibility, we compared the behavior of the two reporters in animals exposed to very low doses of the proteasome inhibitor bortezomib (*Figure 3—figure supplement 1*). Because the effect of bortezomib on proteasome function may be masked by SKN-1A-dependent compensation, we monitored UB(G76V)::GFP levels in both wild type and *skn-1a* mutant animals. We found that very low concentrations of bortezomib (2 ng/ml) cause increased accumulation of UB(G76V)::GFP in *skn-1a* mutant animals. But wild type animals exposed to bortezomib at the same concentration do not show activation of *rpt-3p::gfp*. This suggests that monitoring UB(G76V)::GFP accumulation in a *skn-1a* mutant background serves as a more sensitive indicator of proteasome impairment than *rpt-3p::gfp* expression in wild type animals. As such, the UB(G76V)::GFP reporter should be sensitive enough to detect impairment of proteasome function, if this were the mechanism through which misfolded MHC B or Aβ cause activation of *rpt-3p::gfp*. These results therefore suggest that SKN-1A may be activated by misfolded proteins even in the absence of impaired proteasome function.

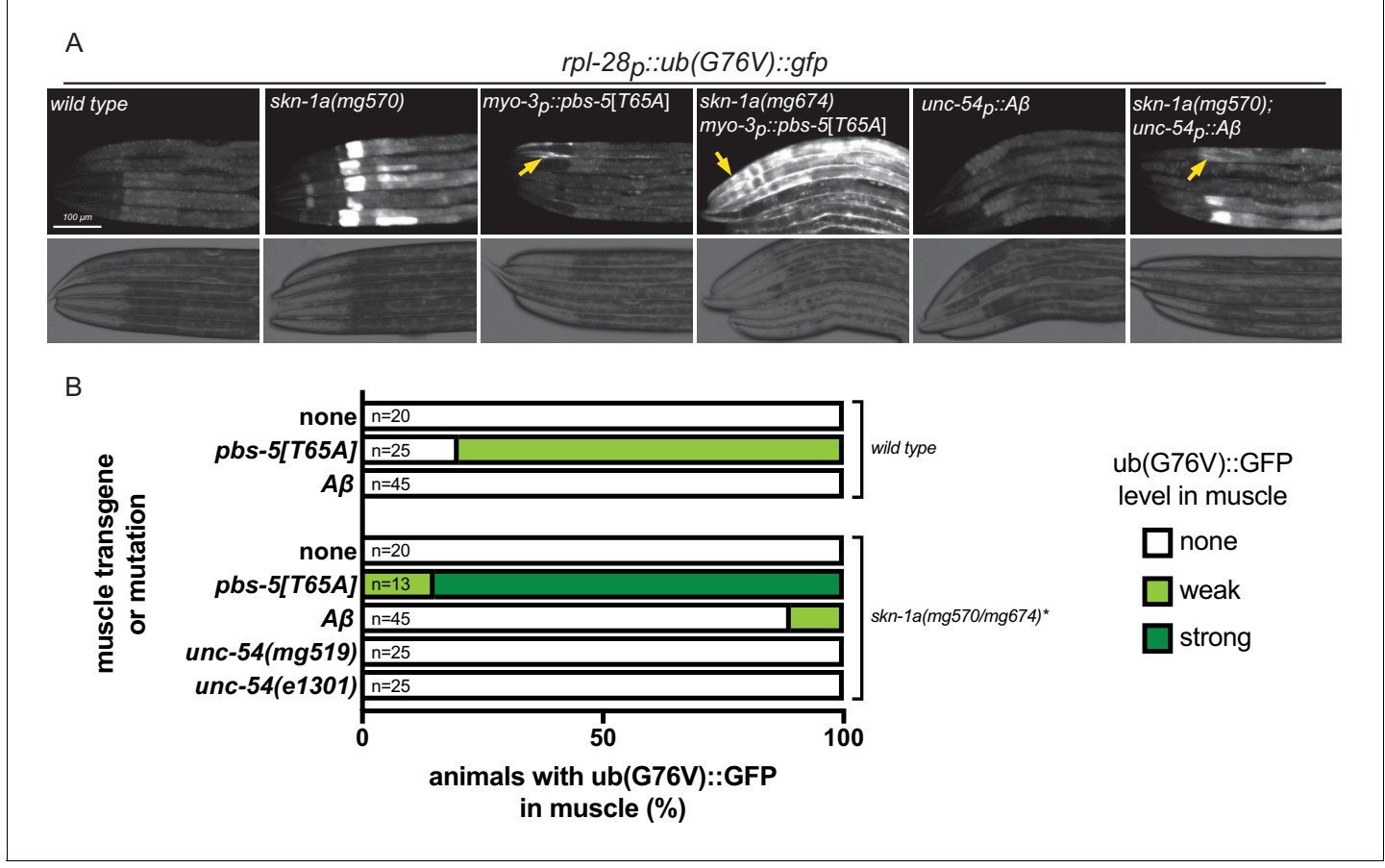

**Figure 3.** Proteasome function is not impaired in animals expressing misfolded proteins. (**a**) Fluorescence micrographs showing impairment of UB (G76V)::GFP degradation in various genotypes. Arrows indicate UB(G76V)::GFP accumulation in muscle cells. (**b**) Comparison of UB(G76V)::GFP stabilization in muscles of animals carrying various SKN-1A-activating transgenes or mutations. *The skn-1a mutation used in the *pbs-5[T65A]* strain is *mg674*, which is an identical CRISPR-induced lesion to *mg570*. All animals were examined for UB(G76V)::GFP stabilization in the muscle at the L4 stage. We note that animals lacking SKN-1A show a defect in basal proteasome function, causing accumulation of UB(G76V)::GFP. This basal effect is limited to the intestine, and so we were still able to detect muscle-specific effects.

DOI: https://doi.org/10.7554/eLife.44425.010

The following figure supplement is available for figure 3:

**Figure supplement 1.** Comparison of *rpt-3p::gfp* activation and UB(G76V)::GFP accumulation in animals exposed to low doses of bortezomib.
DOI: https://doi.org/10.7554/eLife.44425.011

## SKN-1A modulates age-dependent effects of misfolded UNC-54/MHC B

SKN-1A may regulate proteasome capacity to promote clearance of misfolded proteins that may otherwise accumulate and cause cellular dysfunction during aging. If this were the case, we would expect loss of SKN-1A to enhance age-dependent defects in animals expressing misfolded and aggregation-prone proteins. We therefore examined locomotion as a measure of defects in muscle cell function caused by the misfolded proteins that we had identified as activators of SKN-1A. We found no difference in locomotion rate between the wild type and *skn-1a(mg570)* mutants during the first week of adulthood (*Figure 4a*). We measured locomotion of *unc-54(e1301)* and *unc-54 (mg519)* temperature-sensitive myosin heavy chain mutants at 20°C. This condition slows movement but does not cause paralysis of the mutant animals, presumably reflecting partial misfolding of the mutant MHC B. In contrast to wild type, the locomotion of animals harboring *unc-54(e1301)* or *unc-54(mg519)* mutations is strongly modulated by age in a SKN-1A-dependent manner (*Figure 4b,c*). The *unc-54ts* mutants show a severe locomotion defect on day 1 of adulthood, but remarkably, recover to near-normal rates of locomotion on days 3–7. This suggests that during aging the capacity for correct folding and function of mutant MHC B improves. Although age-dependent changes in

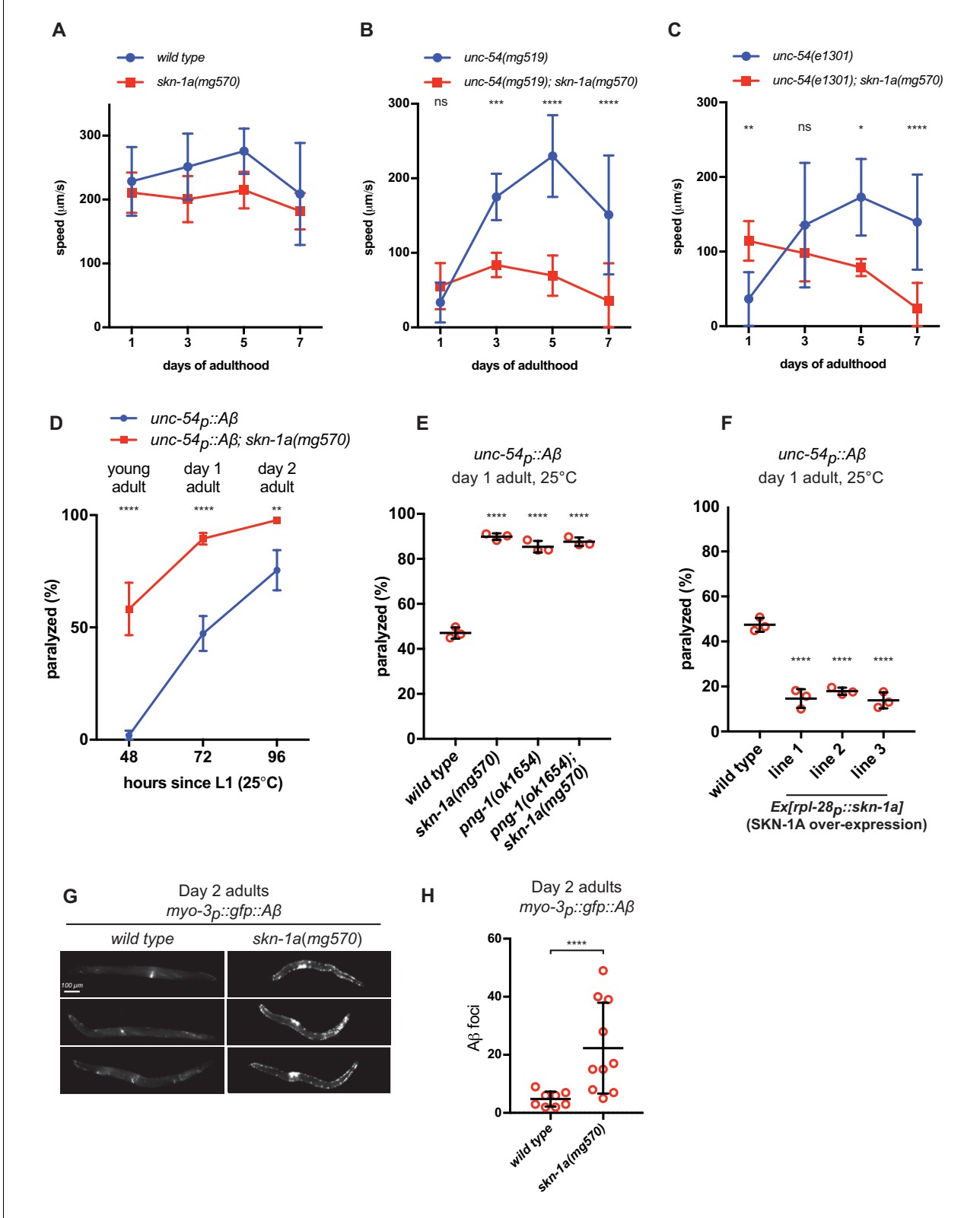

**Figure 4.** SKN-1A ameliorates age-dependent toxicity of misfolded proteins. Analysis of locomotion of (a) wild type and *skn-1a(mg570)* mutant animals, (b) *unc-54(mg519)* and *unc-54(mg519)*; *skn-1a(mg570)* double mutant animals and (c) *unc-54(e1301)* and *unc-54(e1301)*; *skn-1a(mg570)* double mutant animals during aging. (d) Age-dependent paralysis of wild type and *skn-1a(mg570)* mutant Aβ expressing animals. Panels b, c, d: ****$p < 0.0001$; ***$p < 0.001$; **$p < 0.01$; *$p < 0.05$; ns $p > 0.05$ indicates P-value compared to the *skn-1a(+)* control at each time point (two-way ANOVA with Dunnett's

*Figure 4 continued on next page*

*Figure 4 continued*

multiple comparisons test). (e) increased paralysis of Aβ expressing with defective SKN-1A activation. (f) reduced paralysis of Aβ expressing animals with increased SKN-1A levels. Panels e and f: ****p<0.0001 compared to wild type (one-way ANOVA with Tukey's multiple comparisons test). (g) Fluorescence images showing increased accumulation of Aβ::GFP in day two adults in *skn-1a(mg570)* as compared to wild type. (h) Quantification of Aβ::GFP puncta in *wild type* and *skn-1a(mg570)*. ****p<0.0001 (Welch's t-test).

DOI: https://doi.org/10.7554/eLife.44425.012

The following figure supplements are available for figure 4:

**Figure supplement 1.** Activation of *rpt-3p::gfp* in *unc-54ts* mutants is not increased during aging.
DOI: https://doi.org/10.7554/eLife.44425.013
**Figure supplement 2.** The effect of SKN-1A on locomotion of *unc-54ts* mutants on day 1 of adulthood.
DOI: https://doi.org/10.7554/eLife.44425.014

proteostasis and physiology are thought to be largely detrimental, this suggests that in some cases they may include activation of protective responses that improve protein folding or function. Strikingly, this beneficial effect of age is entirely dependent on SKN-1A. *unc-54(mg519); skn-1a(mg570)* double mutants show no age-dependent improvement in locomotion and *unc-54(e1301); skn-1a (mg570)* double mutants show a slight age-dependent decline in locomotion (***Figure 4b,c***). Since two independent *unc-54ts* mutations have the same age-dependent genetic interaction with *skn-1a*, this is not allele-specific, but rather a general effect of SKN-1A on the function of misfolding MHC B. We measured *rpt-3p::gfp* expression in day 1 and day 5 *unc-54(e1301)* and *unc-54(mg519)* mutant adults. Expression of the *rpt-3p::gfp* reporter was unchanged, suggesting that SKN-1A activity does not increase as *unc-54ts* animals age (***Figure 4—figure supplement 1***). Thus, although SKN-1A is needed for *unc-54ts* animals to recover locomotion as they age, this is not caused by age-dependent changes in SKN-1A activity.

Although the rate of movement of *unc-54ts; skn-1a(mg570)* double mutant animals is significantly reduced compared to *unc-54ts* single mutants on later days of adulthood (days 5–7), it is not reduced in day 1 adults. In fact, the locomotor rate of each double mutant is slightly increased compared to the corresponding *unc-54ts* single mutant on day 1 of adulthood (***Figure 4—figure supplement 2***). These data show that activation of the proteasome by SKN-1A is required to maintain muscle function in *unc-54ts* mutant animals as they age, rather than an age-independent requirement for SKN-1A to ensure folding or function of mutant MHC B. SKN-1A is essential for the locomotion of day 1 adults with impaired proteasome function in the muscle (*myo-3p::pbs-5[T65A]* transgenics; ***Figure 2e***), so these data also confirm that mutant MHC B activates SKN-1A without impairing proteasome function as strongly as *myo-3p::pbs-5[T65A]*. Taken together, these results indicate that SKN-1A mediates functionally distinct responses to proteasome dysfunction and expression of misfolded proteins in the muscle. SKN-1A is essential for muscle function during proteasome impairment, regardless of age. In contrast, SKN-1A modulates an age-dependent defect in muscle function caused by misfolded MHC B.

## SKN-1A mitigates accumulation and toxicity of Aβ

Expression of human Aβ in *C. elegans* muscle cells causes progressive adult-onset paralysis (***Link, 2006***). Paralysis is accompanied by aggregation and formation of amyloid fibrils, features also associated with adult-onset neurodegeneration in Alzheimer's disease (***Fay et al., 1998***; ***Link, 1995***). Adult-onset paralysis caused by human Aβ in *C. elegans* muscle is enhanced in *skn-1a(mg570)* mutants (***Figure 4d***). The effects of Aβ are also enhanced in *png-1(ok1654)*, consistent with the failure of the *png-1* mutant to activate SKN-1A (***Figure 4e***). The paralysis of *png-1(ok1654); skn-1a (mg570)* double mutants is not enhanced compared to either single mutant, supporting the model that PNG-1 acts through SKN-1A to mitigate Aβ toxicity. Overexpression of SKN-1A reduces the paralysis caused by muscle-specific Aβ expression in wild type (***Figure 4f***). These data indicate that proteasome activation by SKN-1A is required and sufficient to mitigate the age-dependent toxic effects of Aβ.

Using animals expressing Aβ fused to GFP (*myo-3p::gfp::Aβ*), we monitored expression and localization of Aβ in muscles of wild type and *skn-1a* mutant animals. In day 2 adults, levels of GFP::Aβ were consistently higher in the muscles of *skn-1a* mutant animals than wild type (***Figure 4g***), and *skn-1a* mutant muscles contained many more puncta of localized GFP::Aβ accumulation, suggesting

increased formation of Aβ-containing aggregates (*Figure 4h*). These data suggest that the enhanced adult-onset paralysis in animals that lack SKN-1A is caused by higher levels of Aβ accumulation and aggregation.

## ER-associated SKN-1A promotes longevity and healthy aging

Accumulation of misfolded and aggregated proteins is thought to cause decline in cellular function and health during aging (*David et al., 2010*; *López-Otín et al., 2013*; *Walther et al., 2015*). Mutations that affect both SKN-1A and SKN-1C reduce lifespan, but the individual contribution of SKN-1A is not known (*Blackwell et al., 2015*). We found that *skn-1a(mg570)*, which affects only SKN-1A, causes ~20% reduction in lifespan compared to the wild type (*Figure 5a*). The lifespan of *skn-1a/c (zu67)* animals lacking both SKN-1A and SKN-1C is the same as that of *skn-1a(mg570)* (*Figure 5b*), showing that the effect of *skn-1a/c(zu67)* on lifespan can be explained by loss of SKN-1A. The *mgTi1* [*rpl-28p::skn-1a::gfp*] single copy transgene expresses a functional SKN-1A::GFP fusion protein under the control of the constitutively active *rpl-28* promoter (*Lehrbach and Ruvkun, 2016*). This transgene rescues the bortezomib sensitivity and maternal effect lethality of *skn-1a/c(zu67)* mutants. The lifespan of *skn-1a/c(zu67)*; *mgTi1*[*rpl-28p::skn-1a::gfp*] animals is not reduced compared to wild type, indicating that SKN-1A is sufficient to confer normal lifespan in the absence of SKN-1C. In fact, the lifespan of the rescued animals was reproducibly longer than the wild type (*Figure 5c*). This single copy transgene drives expression from the *rpl-28* ribosome subunit promoter so that SKN-1A::GFP is likely to be overexpressed compared to endogenous SKN-1A. Other independently isolated single-copy *rpl-28p::skn-1a* transgenes also extend lifespan (*Figure 5d*). Thus, SKN-1A is necessary for normal lifespan and sufficient to extend lifespan when over-expressed. Like *skn-1a(mg570)*, the lifespan of *png-1(ok1654)* mutant animals is reduced compared to wild type (*Figure 5e*). *png-1(ok1654)* lifespan is shorter than the *skn-1a(mg570)* mutant, suggesting that PNG-1 might promote longevity through additional SKN-1A-independent pathways. The lifespan of *png-1(ok1654)*; *skn-1a(mg570)* double mutants is not further reduced compared to the *png-1(ok1654)* mutant, indicating that both genes act in the same pathway that controls lifespan (*Figure 5f*). These data suggest that the PNG-1-dependent processing of SKN-1A following release from the ER is required for this transcription factor to regulate lifespan.

Age-dependent defects in vulval integrity are correlated with reduced *C. elegans* lifespan and have been proposed as a marker of healthspan. These defects in vulval integrity are increased by *skn-1(RNAi)*, which depletes multiple SKN-1 isoforms (*Leiser et al., 2016*). *skn-1a(mg570)* and *skn-1a/c(zu67)* animals both show dramatically increased age-dependent vulval integrity defects (*Figure 5g*). This age-dependent vulval degeneration is rescued in *skn-1a/c(zu67)* animals carrying the *mgTi1*[*rpl-28p::skn-1a::gfp*] transgene. Thus, loss of SKN-1A causes the vulval degeneration of *skn-1* mutants. *png-1(ok1654)* mutant animals also show defects in vulval integrity, similar to the *skn-1a(mg570)* mutant (*Figure 5g*). Vulval degeneration is not enhanced in the *png-1(ok1654)*; *skn-1a (mg570)* double mutant, suggesting that both genes act in the same genetic pathway governing vulval integrity during aging. We conclude that regulation of the proteasome by SKN-1A promotes healthy aging and longevity.

## Discussion

We have found that unfolded or aggregated proteins elicit a signal transduced by the SKN-1A/Nrf1 transcription factor, which activates proteasome subunit gene expression. This pathway allows cells to respond to protein folding defects by increasing proteasome levels, enabling more efficient destruction of unfolded or aggregated proteins. We show that this pathway mitigates the age-dependent effects of chronic protein misfolding and aggregation, ensures healthy aging and promotes longevity. Collectively, these data reveal a new unfolded protein response pathway that maintains proteostasis and cellular function during aging (*Figure 6*).

Diverse proteotoxic insults might be expected to engage SKN-1A, however our genetic analyses suggest that this transcription factor responds selectively to cytosolic unfolded proteins and impaired proteasome activity. Proteasome dysfunction can occur as a consequence of oxidative stress, ER stress, and mitochondrial dysfunction (*Bulteau et al., 2001*; *Menéndez-Benito et al., 2005*; *Segref et al., 2014*). But our unbiased genetic analysis of transcriptional regulation of the proteasome thus far has only pointed to mutations that impair the proteasome itself and the misfolding

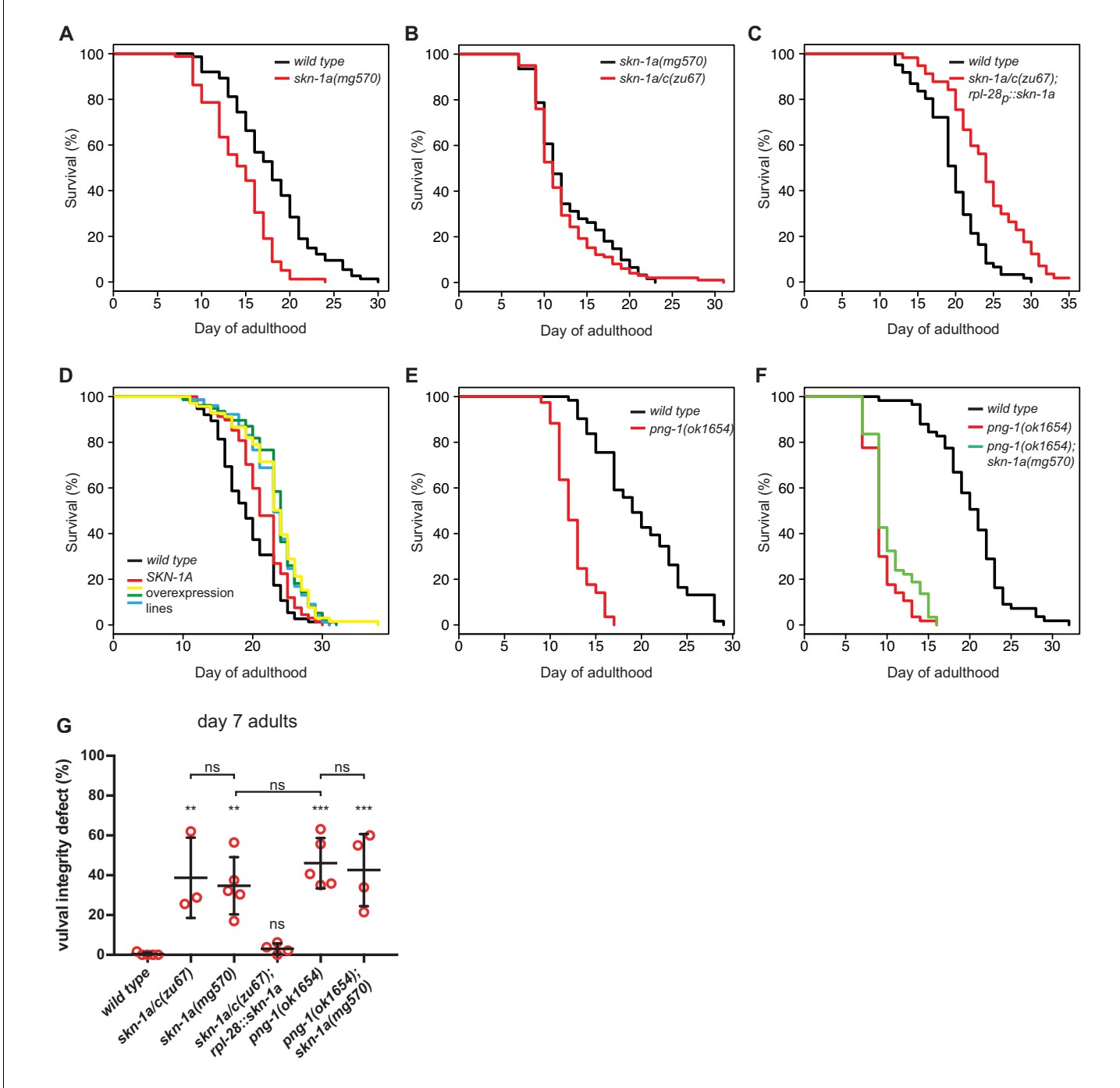

**Figure 5.** SKN-1A and PNG-1 control lifespan. (a–f) Experiments showing that SKN-1A and PNG-1 control lifespan, and that SKN-1A accounts for the effect of *skn-1a/c* mutations on normal lifespan: (a) The lifespan of *skn-1a(mg570)* mutant animals is reduced compared to the wild type. (b) The lifespan of *skn-1a/c(zu67)* mutant animals is not further reduced compared to *skn-1a(mg570)*. (c) The reduced lifespan of *skn-1a/c(zu67)* mutant animals is rescued by a transgene expressing SKN-1A under control of the *rpl-28* promoter. (d) Overexpression of SKN-1A increases lifespan. In five independent *rpl-28p::skn-1a::gfp* lines we found a 10–20% increase in lifespan compared to the wild type. (e) The lifespan of *png-1(ok1654)* mutant animals is reduced compared to wild type. (f) Removal of SKN-1A does not further reduce the lifespan of *png-1(ok1654)* mutant animals. For summary of lifespan statistics see ***Supplementary file 1*** (g) Analysis of vulval degeneration in day 7 adults. ***$p<0.001$; **$p<0.01$; ns $p>0.05$; P-value compared to wild type control is shown unless otherwise indicated (one-way ANOVA with Sidak's multiple comparisons test).

DOI: https://doi.org/10.7554/eLife.44425.015

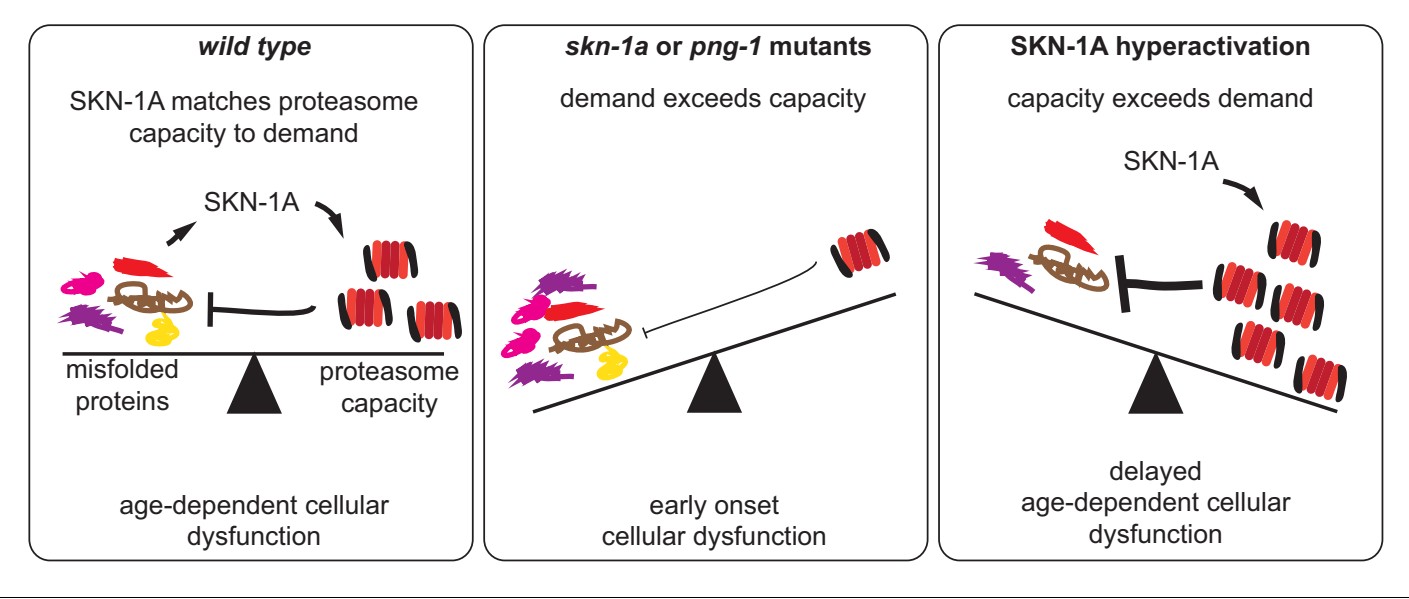

**Figure 6.** SKN-1A modulates functional decline during aging by adjusting proteasome capacity to meet demand for degradation of misfolded proteins. During aging, misfolded proteins eventually accumulate to levels that disrupt cellular function. SKN-1A adjusts proteasome capacity to meet demand for degradation of damaged and misfolded proteins. This modulates the age-dependent accumulation and toxicity of misfolded proteins, thereby altering the rate of functional decline during aging. In animals lacking this pathway (i.e. *skn-1a* or *png-1* mutants), insufficient proteasome capacity leads to a rapid decline and reduced lifespan. Conversely, enhancement of this pathway (by increasing SKN-1A levels or activity) delays the cellular dysfunction caused by misfolded proteins and extends lifespan.

DOI: https://doi.org/10.7554/eLife.44425.016

of a very abundant cytoplasmic protein, UNC-54, as SKN-1A activators. Proteotoxic insults that do not activate SKN-1A/Nrf1 might activate other SKN-1/Nrf isoforms instead; for example oxidative stress activates SKN-1C in *C. elegans* and Nrf2 in mammalian cells. This suggests that the different SKN-1 isoforms – and mammalian Nrfs – have evolved distinct mechanisms of regulation to allow cells to mount appropriate responses to various types of proteotoxic stress.

Our genetic screen yielded multiple *unc-54* alleles rather than a collection of lesions which disrupt the folding of many different proteins. It is possible that only very abundant misfolded proteins will activate *rpt-3*p::*gfp* sufficiently to be detected in this screen. Our screen was also designed to isolate viable mutants. Since many of the most highly expressed proteins perform essential functions, this may have prevented us from isolating mutations that disrupt their folding. Muscle cells show higher levels of both basal and induced *rpt-3*p::*gfp* expression compared to other tissues (*Figure 1—figure supplement 1*). Some combination of the abundance of the UNC-54 protein, its function in muscle, and the viability of *unc-54* mutants may have conspired to make this gene a major target in our screen. An interesting possibility is that the proteasome is particularly important for the regulated degradation of misassembled sarcomeres of the muscle, a tissue that undergoes rapid protein synthesis and turnover – for example during exercise-mediated muscle growth or atrophy during prolonged inactivity.

A detailed elucidation of the mechanism that links accumulation of unfolded proteins to SKN-1A activation will be of great future interest. Our genetic analysis suggests that activation of SKN-1A by misfolded proteins requires release from the ER by ERAD/SEL-1, deglycosylation by PNG-1 and cleavage by DDI-1. These post-translational processing steps are also required for SKN-1A activation during proteasome dysfunction. This suggests that unfolded proteins impair proteasomal degradation of SKN-1A. This model is compatible with the ability of misfolded proteins to cause proteasome dysfunction (*Ayyadevara et al., 2015*; *Bence et al., 2001*; *Gregori et al., 1995*; *Kristiansen et al., 2007*; *Snyder et al., 2003*). However, we detect SKN-1A activation under conditions that have little or no effect on the degradation of a heterologous proteasome substrate. This suggests that proteasome dysfunction is not required for misfolded proteins to trigger SKN-1A activation. One possibility

is that SKN-1A is exquisitely sensitive to changes in proteasome substrate load, and is activated by increased delivery of proteins to the proteasome – even if the increased substrate load does not reach a level that exceeds proteasome capacity. It is also possible that SKN-1A interacts directly with sensor(s) of cellular protein folding that regulate its activity or stability. Interestingly, SKN-1A/Nrf1 itself behaves like an unfolded protein: it is a substrate of the ERAD pathway (*Lehrbach and Ruvkun, 2016*; *Steffen et al., 2010*), which normally functions to eject misfolded glycoproteins from the ER; SKN-1A/Nrf1 activation requires deglycosylation by PNGase, an enzyme that preferentially acts on denatured glycoproteins (*Hirsch et al., 2004*); and Nrf1 is prone to form aggregates in the cytoplasm of cells under proteotoxic stress (*Sha and Goldberg, 2016*). This property could facilitate interactions with cellular sensors of protein folding that may influence SKN-1A/Nrf1 activation. Whatever the mechanism, activation of SKN-1A by misfolded proteins – in the absence of outright proteasome dysfunction – could allow cells to adjust proteasome abundance to meet demand for targeted destruction of damaged or misfolded proteins before they reach levels that compromise cellular function.

Our data are consistent with a model in which SKN-1A boosts various protein quality control pathways that rely on the ubiquitin-proteasome system to eliminate aberrant or damaged proteins. In the case of pathologically misfolding proteins such as Aβ, it is easy to imagine how enhanced elimination of the toxic molecule could limit accumulation over time and so delay the onset of pathology. The explanation for the effects of age and SKN-1A on muscle function in *unc-54ts* mutants must be more complex. SKN-1A is required for the unusual recovery of locomotor function of *unc-54ts* animals that occurs as they age. But SKN-1A activity levels do not change as *unc-54ts* animals get older. This recovery of muscle function is therefore unlikely to be directly mediated by SKN-1A, but likely requires SKN-1A in addition to another unidentified mechanism. It is striking that the function of temperature sensitive mutant proteins, including the mutant MHC B expressed by the *e1301* and *e1157* alleles, is disrupted by the presence of other misfolded or aggregation-prone proteins (*Gidalevitz et al., 2006*; *Olzscha et al., 2011*). By limiting the accumulation of misfolded proteins globally, SKN-1A may create a cellular environment more conducive to the correct folding and function of mutant MHC B.

The accumulation of misfolded and aggregated proteins is a hallmark of aging that has been observed in many species including *C. elegans* (*David et al., 2010*; *Walther et al., 2015*). The effects of SKN-1A and other unfolded protein response pathways on aging and longevity supports the model that protein misfolding and aggregation is a cause rather than a consequence of functional decline during aging (*Denzel et al., 2014*; *Hsu et al., 2003*; *Walker and Lithgow, 2003*). The *skn-1* gene is a component in several *C. elegans* longevity pathways, but the precise mechanism(s) through which *skn-1* promotes longevity are not fully understood (*Blackwell et al., 2015*). The UPR that we have uncovered requires SKN-1A, but not other SKN-1 isoforms, which do not undergo the post-translational modifications necessary for regulation of the proteasome (*Lehrbach and Ruvkun, 2016*). We show that the lifespan and healthspan effects of *skn-1* mutations are largely explained by loss of SKN-1A, and that elevated SKN-1A levels are sufficient to extend lifespan, even in animals that lack SKN-1C. Thus our data suggests that the *skn-1* gene primarily promotes longevity by safeguarding proteostasis through SKN-1A/Nrf1-dependent control of proteasome expression and activity.

The failure of proteasome-dependent protein quality control systems is intimately linked to neurodegeneration. Intracellular inclusions that contain ubiquitinated proteins are a central feature of essentially all neurodegenerative diseases (*Alves-Rodrigues et al., 1998*). Depletion of Nrf1 in the mouse brain causes neurodegeneration accompanied by formation of ubiquitin-containing inclusions in young animals (*Lee et al., 2011*). A recent study has suggested that pharmacological activation of Nrf1 is protective in a mouse model of one age-dependent neurodegenerative condition – spinal and bulbar muscular atrophy (*Bott et al., 2016*), and our data indicates SKN-1A/Nrf1 is similarly protective in a *C. elegans* model of Alzheimer's disease. We therefore suggest that increasing the activity of Nrf1 may be beneficial for human aging and treatment of various adult-onset neurodegenerative diseases.

# Materials and methods

## Key resources table

| Reagent type (species) or resource | Designation | Source or reference | Identifiers | Additional information |
|---|---|---|---|---|
| Strain, strain background (*E. coli*) | *E. coli* OP50 | CGC | OP50 | |
| Strain, strain background (*E. coli*) | *E. coli* HT115 | CGC | HT115 | |
| Strain, strain background (*C. elegans*) | *unc-54(e1301) I.* | CGC | CB1301 | |
| Strain, strain background (*C. elegans*) | *dvIs2* | CGC | CL2006 | unc-54::Aβ |
| Strain, strain background (*C. elegans*) | *dvIs37* | CGC | CL2331 | myo-3::gfp::Aβ |
| Strain, strain background (*C. elegans*) | *mgIs72 II* | *Lehrbach and Ruvkun, 2016* | GR2183 | rpt-3::gfp integrated array |
| Strain, strain background (*C. elegans*) | *pbs-5(mg502) I; mgIs72 II* | *Lehrbach and Ruvkun, 2016* | GR2184 | proteasome mutant |
| Strain, strain background (*C. elegans*) | *mgIs72 II; skn-1(mg570) IV* | *Lehrbach and Ruvkun, 2016* | GR2197 | |
| Strain, strain background (*C. elegans*) | *mgIs72 II; ddi-1(mg571) IV* | *Lehrbach and Ruvkun, 2016* | GR2211 | |
| Strain, strain background (*C. elegans*) | *unc-119(ed3) III; mgTi4* | *Lehrbach and Ruvkun, 2016* | GR2212 | rpl-28::ha::skn-1a::gfp::tbb-2 |
| Strain, strain background (*C. elegans*) | *unc-119(ed3) III; mgTi5* | *Lehrbach and Ruvkun, 2016* | GR2213 | rpl-28::ha::skn-1a::gfp::tbb-2 |
| Strain, strain background (*C. elegans*) | *mgIs72 II; sel-1(mg547) V* | *Lehrbach and Ruvkun, 2016* | GR2215 | Strain, strain |
| background (*C. elegans*) | *unc-119(ed3) III; skn-1(zu67) IV; mgTi1* | *Lehrbach and Ruvkun, 2016* | GR2221 | rpl-28::skn-1a::GFP::tbb-2 rescues skn-1(zu67) |
| Strain, strain background (*C. elegans*) | *png-1(ok1654) I; mgIs72 II* | *Lehrbach and Ruvkun, 2016* | GR2236 | |
| Strain, strain background (*C. elegans*) | *skn-1(mg570) IV* | *Lehrbach and Ruvkun, 2016* | GR2245 | |
| Strain, strain background (*C. elegans*) | *png-1(ok1654) I* | CGC | GR2246 | |
| Strain, strain background (*C. elegans*) | *png-1(ok1654) I; skn-1(mg570) IV* | this study | GR3089 | Reagent requests: see Materials and methods |
| Strain, strain background (*C. elegans*) | *mgIs77 V* | this study | GR3090 | rpl-28::ub(G76V)::gfp::tbb-2, myo-3::mcherry marker. Reagent requests: see Materials and methods |
| Strain, strain background (*C. elegans*) | *unc-119(ed3) III; mgTi15* | this study | GR3091 | rpl-28::skn-1a::GFP::tbb-2. Reagent requests: see Materials and methods |
| Strain, strain background (*C. elegans*) | *unc-119(ed3) III; mgTi17* | this study | GR3092 | rpl-28::HA::skn-1a::GFP::tbb-2. Reagent requests: see Materials and methods |
| Strain, strain background (*C. elegans*) | *skn-1(mg570) IV; mgIs77 V* | this study | GR3094 | rpl-28::ub(G76V)::gfp::tbb-2. Reagent requests: see Materials and methods |

*Continued on next page*

*Continued*

| Reagent type (species) or resource | Designation | Source or reference | Identifiers | Additional information |
|---|---|---|---|---|
| Strain, strain background (*C. elegans*) | *mgIs72 II; pas-1(mg511) V* | this study | GR3141 | proteasome mutant. Reagent requests: see Materials and methods |
| Strain, strain background (*C. elegans*) | *rpn-10(mg525) I; mgIs72 II* | this study | GR3142 | proteasome mutant. Reagent requests: see Materials and methods |
| Strain, strain background (*C. elegans*) | *mgIs72 II; rpn-1(mg514) IV* | this study | GR3143 | proteasome mutant. Reagent requests: see Materials and methods |
| Strain, strain background (*C. elegans*) | *pbs-5(mg509) I; mgIs72 II* | this study | GR3144 | proteasome mutant. Reagent requests: see Materials and methods |
| Strain, strain background (*C. elegans*) | *mgIs72 II; rpt-6(mg513) III* | this study | GR3145 | proteasome mutant. Reagent requests: see Materials and methods |
| Strain, strain background (*C. elegans*) | *rpn-10(mg495) I; mgIs72 II* | this study | GR3146 | proteasome mutant. Reagent requests: see Materials and methods |
| Strain, strain background (*C. elegans*) | *mgIs72 II; rpt-6(mg512) III* | this study | GR3147 | proteasome mutant. Reagent requests: see Materials and methods |
| Strain, strain background (*C. elegans*) | *mgIs78 IV* | this study | GR3148 | myo-3::H2B::mcherry::SL2::pbs-5[T65A] (pNL47). Reagent requests: see Materials and methods |
| Strain, strain background (*C. elegans*) | *mgIs72 II; mgIs78 IV* | this study | GR3149 | Reagent requests: see Materials and methods |
| Strain, strain background (*C. elegans*) | *rpn-10(mg529) I; mgIs72 II* | this study | GR3150 | proteasome mutant. Reagent requests: see Materials and methods |
| Strain, strain background (*C. elegans*) | *pbs-2(mg530) I; mgIs72 II* | this study | GR3151 | proteasome mutant. Reagent requests: see Materials and methods |
| Strain, strain background (*C. elegans*) | *rpn-10(mg531) I; mgIs72 II* | this study | GR3152 | proteasome mutant. Reagent requests: see Materials and methods |
| Strain, strain background (*C. elegans*) | *unc-54(e190) I; mgIs72 II* | this study | GR3153 | Reagent requests: see Materials and methods |
| Strain, strain background (*C. elegans*) | *mgIs78 IV; mgIs77 V* | this study | GR3154 | myo-3::H2B::mcherry::SL2::pbs-5[T65A] and Ub(G76V)::gfp. Reagent requests: see Materials and methods |
| Strain, strain background (*C. elegans*) | *rpn-11(mg494) mgIs72 II* | this study | GR3155 | proteasome mutant. Reagent requests: see Materials and methods |
| Strain, strain background (*C. elegans*) | *unc-54(mg519) I; mgIs72 II* | this study | GR3156 | *unc-54*ts. Reagent requests: see Materials and methods |
| Strain, strain background (*C. elegans*) | *unc-54(mg519) I* | this study | GR3157 | *unc-54*ts. Reagent requests: see Materials and methods |
| Strain, strain background (*C. elegans*) | *mgIs72 II; skn-1 (mg674) mgIs78 IV* | this study | GR3158 | mg674 causes G2STOP in SKN-1A. Reagent requests: see Materials and methods |
| Strain, strain background (*C. elegans*) | *unc-54(e1157) I; mgIs72 II* | this study | GR3159 | *unc-54*ts. Reagent requests: see Materials and methods |

*Continued*

| Reagent type (species) or resource | Designation | Source or reference | Identifiers | Additional information |
|---|---|---|---|---|
| Strain, strain background (*C. elegans*) | unc-54(e1301) I; mgIs72 II | this study | GR3160 | *unc-54*ts. Reagent requests: see Materials and methods |
| Strain, strain background (*C. elegans*) | unc-54(mg528) I; mgIs72 II | this study | GR3161 | *unc-54*ts. Reagent requests: see Materials and methods |
| Strain, strain background (*C. elegans*) | unc-54(mg540) I; mgIs72 II | this study | GR3162 | *unc-54*ts. Reagent requests: see Materials and methods |
| Strain, strain background (*C. elegans*) | skn-1(mg674) mgIs78 IV | this study | GR3163 | mg674 causes G2STOP in SKN-1A. Reagent requests: see Materials and methods |
| Strain, strain background (*C. elegans*) | unc-54(e1301) I; mgIs72 II; skn-1(mg570) IV | this study | GR3164 | Reagent requests: see Materials and methods. |
| Strain, strain background (*C. elegans*) | unc-54(e1301) I; skn-1(mg570) IV | this study | GR3165 | Reagent requests: see Materials and methods |
| Strain, strain background (*C. elegans*) | unc-54(mg519) I; mgIs72 II; skn-1(mg570) IV | this study | GR3166 | Reagent requests: see Materials and methods |
| Strain, strain background (*C. elegans*) | unc-54(mg519) I; skn-1(mg570) IV | this study | GR3167 | Reagent requests: see Materials and methods |
| Strain, strain background (*C. elegans*) | skn-1(mg674) mgIs78/nT1[qIs51] IV; mgIs77/nT1[qIs51] V | this study | GR3168 | skn-1(mg674) mgIs78; mgIs77 animals are very sick, use balancer to maintain. Reagent requests: see Materials and methods |
| Strain, strain background (*C. elegans*) | unc-54(e1301) I; skn-1(mg570) IV; mgIs77 V | this study | GR3169 | Reagent requests: see Materials and methods |
| Strain, strain background (*C. elegans*) | unc-54(mg519) I; skn-1(mg570) IV; mgIs77 V | this study | GR3170 | Reagent requests: see Materials and methods |
| Strain, strain background (*C. elegans*) | pbs-3(mg527) mgIs72 II | this study | GR3171 | proteasome mutant. Reagent requests: see Materials and methods |
| Strain, strain background (*C. elegans*) | pbs-2(mg581) I; mgIs72 II | this study | GR3172 | proteasome mutant. Reagent requests: see Materials and methods |
| Strain, strain background (*C. elegans*) | rpn-9(mg533) mgIs72 II | this study | GR3173 | proteasome mutant. Reagent requests: see Materials and methods |
| Strain, strain background (*C. elegans*) | rpn-8(mg587) I; mgIs72 II | this study | GR3174 | proteasome mutant. Reagent requests: see Materials and methods |
| Strain, strain background (*C. elegans*) | rpn-5(mg534) mgIs72 II | this study | GR3175 | proteasome mutant. Reagent requests: see Materials and methods |
| Strain, strain background (*C. elegans*) | rpn-8(mg536) I; mgIs72 II | this study | GR3176 | proteasome mutant. Reagent requests: see Materials and methods |
| Strain, strain background (*C. elegans*) | mgIs72 II; rpn-1(mg537) IV | this study | GR3177 | proteasome mutant. Reagent requests: see Materials and methods |
| Strain, strain background (*C. elegans*) | pbs-2(mg538) I; mgIs72 II | this study | GR3178 | proteasome mutant. Reagent requests: see Materials and methods |
| Strain, strain background (*C. elegans*) | pbs-4(mg539) I; mgIs72 II | this study | GR3179 | proteasome mutant. Reagent requests: see Materials and methods |

*Continued on next page*

*Continued*

| Reagent type (species) or resource | Designation | Source or reference | Identifiers | Additional information |
|---|---|---|---|---|
| Strain, strain background (*C. elegans*) | *mgIs72 II; dvIs2* | this study | GR3180 | Amyloid beta + rpt-3::gfp. Reagent requests: see Materials and methods |
| Strain, strain background (*C. elegans*) | *mgIs72 II; dvIs2; skn-1(mg570) IV* | this study | GR3181 | Amyloid beta + rpt-3::gfp in *skn-1a* mutant. Reagent requests: see Materials and methods |
| Strain, strain background (*C. elegans*) | *skn-1(mg570) IV; mgIs77 V; dvIs2* | this study | GR3182 | unc-54::Aβ+Ub(G76V):: gfp in *skn-1a* mutant. Reagent requests: see Materials and methods |
| Strain, strain background (*C. elegans*) | *mgIs77 V; dvIs2* | this study | GR3183 | unc-54::Aβ+Ub(G76V)::gfp. Reagent requests: s ee Materials and methods |
| Strain, strain background (*C. elegans*) | *skn-1(mg570) IV; dvIs2* | this study | GR3184 | unc-54::Aβ in *skn-1a* mutant. Reagent requests: see Materials and methods |
| strain, strain background (*C. elegans*) | *skn-1(mg570) IV; dvIs37* | this study | GR3185 | myo-3::gfp::Aβ in *skn-1a* mutant. Reagent requests: see Materials and methods |
| Strain, strain background (*C. elegans*) | *png-1(ok1654) I; dvIs2* | this study | GR3186 | unc-54::Aβ in a *png-1* mutant. Reagent requests: see Materials and methods |
| Strain, strain background (*C. elegans*) | *png-1(ok1654) I; skn-1(mg570) IV; dvIs2* | this study | GR3187 | unc-54::Aβ in *png-1 skn-1a* double mutant. Reagent requests: see Materials and methods |
| Strain, strain background (*C. elegans*) | *mgIs72 II; ddi-1(mg571) IV; dvIs2* | this study | GR3188 | unc-54::Aβ in *ddi-1* mutant + rpt-3::gfp. Reagent requests: see Materials and methods |
| Strain, strain background (*C. elegans*) | *png-1(ok1645) I; mgIs72 II; dvIs2* | this study | GR3189 | unc-54::Aβ in *png-1* mutant + rpt-3::gfp. Reagent requests: see Materials and methods |
| Strain, strain background (*C. elegans*) | *dvIs2; mgEx813* | this study | GR3190 | skn-1a overexpression (pNL214), array marked by myo-2::mcherry. Reagent requests: see Materials and methods |
| Strain, strain background (*C. elegans*) | *dvIs2; mgEx814* | this study | GR3191 | skn-1a overexpression (pNL214), array marked by myo-2::mcherry. Reagent requests: see Materials and methods |
| Strain, strain background (*C. elegans*) | *dvIs2; mgEx815* | this study | GR3192 | skn-1a overexpression (pNL214), array marked by myo-2::mcherry. Reagent requests: see Materials and methods |
| Strain, strain background (*C. elegans*) | *mgIs72 II; sel-1(mg547) V; dvIs2* | this study | GR3193 | unc-54::Aβ in *sel-1* mutant + rpt-3::gfp. Reagent requests: see Materials and methods |
| Strain, strain background (*C. elegans*) | *hsf-1(sy441) I; mgIs72* | this study | GR3291 | rpt-3::gfp, *hif-1* mutant. Reagent requests: see Materials and methods |
| Strain, strain background (*C. elegans*) | *unc-119(ed3) III; mgEx831* | this study | GR3292 | rpl-28p::skn-1a[ΔDBD]:: gfp marked by myo-2::mcherry and unc-119(+). Reagent requests: s ee Materials and methods |

*Continued on next page*

*Continued*

| Reagent type (species) or resource | Designation | Source or reference | Identifiers | Additional information |
|---|---|---|---|---|
| Strain, strain background (*C. elegans*) | *unc-54(e1301) I; mgEx831* | this study | GR3293 | rpl-28p::skn-1a[ΔDBD]:: gfp, *unc-54ts* mutant. Reagent requests: see Materials and methods |
| Strain, strain background (*C. elegans*) | *unc-54(mg519) I; mgEx831* | this study | GR3294 | rpl-28p::skn-1a[ΔDBD]::gfp, *unc-54ts* mutant. Reagent requests: see Materials and methods |
| Strain, strain background (*C. elegans*) | *hsf-1(sy441) I; mgIs72; skn-1a(mg570)* | this study | GR3295 | rpt-3::gfp, *hif-1, skn-1a* double mutant. Reagent requests: see Materials and methods |
| Strain, strain background (*C. elegans*) | *skn-1(zu67) IV/nT1 [unc-?(n754) let-?](IV;V)* | CGC | EU1 | |
| Strain, strain background (*C. elegans*) | *wild type* | CGC | N2 | |
| Recombinant DNA reagent (plasmid) | rpl-28::skn-1a::tbb-2 | *Lehrbach and Ruvkun, 2016*. | pNL214 | Reagent requests: see Materials and methods |
| Recombinant DNA reagent (plasmid) | myo-3::mcherry::his-58:: SL2::pbs-5[T65A] | this study | pNL47 | Reagent requests: see Materials and methods |
| Recombinant DNA reagent (plasmid) | rpl-28::ub(G76V)::gfp::tbb-2 | this study | pNL121 | Reagent requests: see Materials and methods |
| Chemical compound, drug | Bortezomib | L C Laboratories | Cat#B1408 | |
| Software, algorithm | ImageJ | NIH | | https://imagej.nih.gov/ij/ |
| Software, algorithm | Zen | Zeiss | | https://www.zeiss.com/ microscopy/us/products/ microscope-software/zen.html |
| Software, algorithm | Ape (A plasmid editor) | M Wayne Davis | | http://jorgensen. biology.utah.edu /wayned/ape/ |
| Software, algorithm | Graphpad Prism | Graphpad | | https://www.graphpad .com/scientific- software/prism/ |

### *C. elegans* maintenance and genetics

*C. elegans* were maintained on standard media at 20°C (unless otherwise indicated) and fed *E. coli* OP50. A list of strains used in this study is provided in the Key Resources Table. RNAi was performed as described in *Kamath and Ahringer (2003)*. Mutagenesis was performed by treatment of L4 animals in 47 mM EMS for 4 hr at 20°C. Some strains were provided by the CGC, which is funded by NIH Office of Research Infrastructure Programs (P40 OD010440). *png-1(ok1654)* was generated by the *C. elegans* Gene Knockout Project at the Oklahoma Medical Research Foundation, part of the International *C. elegans* Gene Knockout Consortium.

### Identification of EMS induced mutations by whole genome sequencing

Genomic DNA was prepared using the Gentra Puregene Tissue kit (Qiagen, #158689) according to the manufacturer's instructions. Genomic DNA libraries were prepared using the NEBNext genomic DNA library construction kit (New England Biololabs, #E6040), and sequenced on a Illumina Hiseq instrument. Deep sequencing reads were analyzed using Cloudmap (*Minevich et al., 2012*).

### Transgenesis

Cloning was performed by isothermal/Gibson assembly (*Gibson et al., 2009*). All plasmids used for transgenesis are listed in the Key Resources Table. All constructs were assembled in pNL43 (*Lehrbach and Ruvkun, 2016*) or in pBluescript. The SKN-1 constructs used in this study are

described in *Lehrbach and Ruvkun (2016)*. Extra-chromosomal arrays were generated using *myo-2:: mcherry* as a co-injection marker. EMS mutagenesis was used to induce integration of extrachromosomal arrays. The *myo-3p::pbs-5[T65A]* construct was generated to expresses mcherry:: histone (H2B) and mutant PBS-5 from an artificial operon under control of the *myo-3* promoter, which drives expression specifically in the body wall muscle (*myo-3p::mcherry::H2B::SL2::PBS-5[T65A]*). The *mcherry::H2B* serves to confirm the tissue specific expression of the transgene. A DNA fragment containing the 5'UTR, coding sequence and 3'UTR of *pbs-5* was cloned and site-directed mutagenesis was used to introduce the T65A mutation. The altered *pbs-5* DNA fragment was then cloned into pBluescript with the *myo-3* promoter (a 2169 bp fragment immediately upstream of the *myo-3* start codon) and mcherry fused in-frame to the *his-58* (H2B) coding sequence (a 373 bp fragment containing the *his-58* open reading frame). The *ub(G76V)::gfp* construct was generated to drive ubiquitous expression of UB(G76V)::GFP under control of the *rpl-28* promoter. A synthesized DNA fragment encoding ubiquitin was cloned in frame with GFP to generate the UB(G76V)::GFP coding sequence. The G76V mutation was introduced by the oligos used for Gibson assembly. This was inserted into pNL43 with the *rpl-28* promoter (605 bp immediately upstream of the *rpl-28* start codon) and tbb-2 3'UTR (376 bp immediately downstream of the *tbb-2* stop codon).

## Genome modification by CRISPR/Cas9

The *mgIs78[myo-3p::mcherry::H2B::SL2::PBS-5[T65A]]* transgene is integrated within chromosome IV and appears to be tightly linked to *skn-1*. The *skn-1a(mg674)* allele is identical to *mg570* and was generated as described in *Lehrbach and Ruvkun (2016)* using *dpy-10(cn64)* as a co-CRISPR marker by injection of *mgIs78* transgenic animals.

## Microscopy

For *rpt-3p::gfp*, *rpl-28p::Ub(G76V)::gfp* and *myo-3p::gfp::Aβ* transgenics, bright field and GFP fluorescence images were collected using a Zeiss AxioZoom V16, equipped with a Hammamatsu Orca flash 4.0 digital camera camera, and using Axiovision software. For *rpl-28p::skn-1a[ΔDBD]::gfp*, DIC and GFP fluorescence images were collected using a Zeiss Axio Image Z1 microscope, equipped with a Zeiss AxioCam HRc digital camera, using Axiovision software. Images were processed using ImageJ software. For all fluorescence images, images shown within the same figure panel were collected using the same exposure time and then processed identically in ImageJ. To quantify *rpt-3p:: gfp* expression, the maximum pixel intensity within a transverse section approximately 25 µm posterior to the pharynx of adult animals was measured using imageJ. To quantify UB(G76V)::GFP stabilization in muscle, images of transgenic animals were manually inspected in imageJ. Weak stabilization was recorded if animals contained low but detectable levels of UB(G76V)::GFP in any part of the body wall muscle (16-bit pixel intensity greater than 500). Strong stabilization was recorded if animals contained higher levels of UB(G76V)::GFP in any part of the body wall muscle (16-bit pixel intensity greater than 2000). Aβ foci were counted using the find maxima tool in imageJ.

## Bortezomib treatment for imaging

Plates were supplemented with bortezomib (LC Laboratories #B1408) by spotting a bortezomib solution on top of NGM plates seeded with OP50. The bortezomib solution was allowed to dry into the plate before adding L4 stage animals. These animals were allowed to reproduce, and reporter expression was imaged in the next generation. All treatment conditions contained less than 0.001% DMSO and bortezomib treated worms were compared to DMSO-treated control animals.

## Sterility assay

L4 animals were selected from mixed stage cultures that had been maintained without starvation for at least two generations and shifted to 20°C or 25°C. In the next generation, L4 animals were picked individually to fresh plates and returned to the same temperature. The production of progeny was monitored over the following 5 days. Animals that produced no progeny were recorded as sterile, all other animals (regardless of brood size or viability of progeny) were recorded as fertile. Fertility of at least 10 animals was assessed for each strain at each temperature. All strains used in fertility assays contained the *mgIs72* transgene.

## Aβ-induced paralysis assay

For each assay at least 100 starvation-synchronized L1 stage animals were raised at 25°C. Animals grown under this condition reach adulthood after ~48 hr. Starting at 48 hr, animals were scored for paralysis every 24 hr. Animals were scored as paralyzed if they showed no sign of movement after tapping the plate or gently prodding the animal.

## unc-54ts paralysis assay

L4 animals were selected from mixed stage cultures that had been maintained without starvation for at least two generations and shifted to 20°C or 25°C. When the majority of the progeny had reached adulthood, adult animals were scored for paralysis. Animals were scored as paralyzed if they showed no sign of movement after tapping the plate or gently prodding the animal. At least 100 animals for each strain under each condition were scored.

## Measurement of locomotor rate (speed)

Locomotor assays were initiated by selecting L4 animals from mixed stage cultures that had been maintained without starvation for at least two generations. L4 animals were maintained for a further 24 hr to assay day one adults, or for correspondingly longer periods to assay day 3, 5 and 7 adults. For assays in which locomotion was measured on multiple days, a single population of animals was maintained and repeatedly tested. Animals that had bagged or ruptured were removed from analysis since these defects impair locomotion but do not reflect changes in body wall muscle function. To assay locomotor rate, each animal was transferred to a fresh plate seeded with OP50 and then removed after 1 min. An image of the tracks left in the lawn by each animal was collected. The distanced travelled was then measured using imageJ and used to calculate average speed.

## Lifespan analysis

Lifespan assays were initiated by selecting L4 animals from mixed stage cultures that had been maintained without starvation for at least two generations. Animals were transferred to fresh plates on day three and then every 2 days until reproduction ceased and every 3–5 days thereafter. Animals were checked for survival at least every other day. Animals that died by bagging or crawling off the plates were censored. Animals that died due to age-related vulval integrity defects (after ceasing reproduction, when such defects can be distinguished from bagging) were not censored from analysis, as this is a major mode of age-dependent lethality of some of the mutants analyzed. Survival curves, calculation of mean lifespan and statistical analysis was performed in R using the 'survival' package. The log-rank (Mantel-Haenszel) test was used to compare survival curves. Statistics for all assays (including replicate assays not shown in main figures) are shown in *Supplementary file 1*.

## Scoring of age-related vulval integrity defects

Assays to measure age-related vulval integrity defects were initiated by selecting L4 animals from mixed-stage cultures that had been maintained without starvation for at least two generations. Animals were transferred to fresh plates on days 3 and 5 of the assay. On days 5 and 7, animals were checked for rupture, and the cumulative total number of animals ruptured during the first week of adulthood recorded. 30–80 animals were analyzed in each assay. At least three replicate assays were performed for each genotype.

## Statistical analysis

Statistical analyses of lifespan data are described in the lifespan analysis section. All other statistical analyses were performed using Graphpad Prism. All biological replicates were performed with independent populations of animals.

## Additional information

### Funding

| Funder | Grant reference number | Author |
|---|---|---|
| Grace Science Foundation | | Nicolas J Lehrbach<br>Gary Ruvkun |
| National Institutes of Health | R01 AG016636 | Gary Ruvkun |

The funders had no role in study design, data collection and interpretation, or the decision to submit the work for publication.

### Author contributions

Nicolas J Lehrbach, Conceptualization, Funding acquisition, Validation, Investigation, Visualization, Methodology, Writing—original draft, Writing—review and editing; Gary Ruvkun, Conceptualization, Supervision, Funding acquisition, Writing—original draft, Writing—review and editing

### Author ORCIDs

Nicolas J Lehrbach (iD) http://orcid.org/0000-0001-7342-4136
Gary Ruvkun (iD) http://orcid.org/0000-0002-7473-8484

### Decision letter and Author response

Decision letter https://doi.org/10.7554/eLife.44425.020
Author response https://doi.org/10.7554/eLife.44425.021

## Additional files

### Supplementary files

• Supplementary file 1. Lifespan data and statistics.
DOI: https://doi.org/10.7554/eLife.44425.017

### Data availability

All data analyzed or generated in this study are included in the figures and supporting files.

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
