## [Decision Letter]

Thank you for submitting your article "ER-associated SKN-1A/Nrf1 mediates a cytoplasmic unfolded protein response" for consideration by *eLife*. Your article has been reviewed by four peer reviewers, one of whom is a member of our Board of Reviewing Editors, and the evaluation has been overseen by a Reviewing Editor and Kevin Struhl as the Senior Editor. The following individuals involved in review of your submission have agreed to reveal their identity: Andrew Dillin (Reviewer #1).

The reviewers have discussed the reviews with one another and the Reviewing Editor has drafted this decision to help you prepare a revised submission.

Summary:

This study reports on a genetic analysis of the role of unfolded cytosolic proteins in promoting Skn-1a/Nrf1-mediated activation of genes encoding proteasomal subunits. Its two key conclusions are that unfolded cytosolic proteins activate Skn-1a by a mechanism that does not require impairment of proteasome capacity and that this Skn-1a-mediated gene expression programme is important in buffering some of the phenotypic consequences of cytosolic protein misfolding.

The first conclusion rests on the demonstration that a Skn-1a target gene, rpt-3p::gfp, is activated by (genetic manipulations that induce) cytosolic protein misfolding in the absence of evidence for proteasomal dysfunction (as gaged by the stabilisation of a proteasome reserve indicator UbG76V::GFP).

The second conclusion rests on evidence for an interaction between the skn-1a genotype of animals (wildtype or null) and the phenotypic consequences of triggering misfolding of the product of a temperature sensitive allele of unc-54, a body wall myosin.

Essential revisions:

Please consider the following 4 points in the revised version of the manuscript:

1) Reviewer 2 points out that potential limitations to the dynamic range of the UbG76V::GFP reporter may compromise the strength of the conclusions drawn. The authors convincingly show that a mutation in the catalytic site of one of the catalytic proteasome subunits (pbs-5[T65A]) strongly stabilises the UbG76V::GFP reporter and that the cytosolic protein misfolding-inducing mutations that activate the Skn-1a target gene, rpt-3p::gfp do not share this feature (Figure 3). However, the concern remains that a significant deficit in proteasome capacity in the cytosolic protein misfolding-inducing mutations, might have been missed if the deficit fell under the threshold of the reporter's sensitivity. This concern arises because it is known that ubiquitin-dependent substrates (such as UbG76V::GFP) accumulate only when proteasome activity (as assessed by Suc-LLVY-AMC hydrolysis in cell lysates) is substantially blocked [see Figure 4a, from Dantuma et al., (2000))]. Therefore, it would be very reassuring to learn of a positive correlation between level of proteasome dysfunction, UbG76V::GFP reporter activity and Skn-1a target gene rpt-3p::gfp expression and then to learn that cytosolic protein misfolding is able to activate the Skn-1a target gene rpt-3p::gfp to similar levels without activation of UbG76V::GFP reporter activity. In other words that over a series of mutations/alleles, rpt-3p::gfp expression is correlated with UbG76V::GFP reporter activity when wrought by proteasome dysfunction but dissociated from UbG76V::GFP reporter activity, when wrought by cytosolic protein misfolding. RNAi, proteasome inhibitors or perhaps the numerous ts alleles in proteasome subunits isolated here (Figure 1—figure supplement 1) may be harnessed for this calibration (see point 1 of reviewer 2).

2) The evidence for Skn-1a activation by cytosolic misfolding presently rests on the Skn-1a genotype dependence of rpt-3p::gfp activity (Figure 1D,E). It would be helpful to buttress this convincing genetic argument with evidence that Skn-1a is activated at the protein level. Andy Dillin (reviewer 1) suggests this might be achieved by studying animals with skn-1a::GFP reporter (strain GR2198).

3) The implicit claim that hsf-1∆-dependent activation of rpt-3p::GFP is Skn-1a-dependent (Figure S4) should be supported by evidence that such activation is blocked by a skn-1a deletion.

4) The curious time course of the motility defect in the unc-54 mutant worms, suggests that correlating this phenotype with the temporal profile of Skn-1a activity might provide important support for the hypothesis that Skn-1a buffers the consequences of the misfolding-prone mutant. Therefore, we ask that you consider the specific suggestion of reviewer three and measure the time-dependent changes in Skn-1a activity in unc-54 mutant worms.

Title:

Please spell out "ER" [endoplasmic reticulum]

Reviewer #1:

The manuscript by Lehrbach and Ruvkin presents data to link activation of the ER-resident transcription factor SKN-1A to cytosolic protein misfolding. Ectopic expression of aggregation prone amyloid beta or myosin in addition to mutation of heat shock factor, to activate expression of SKN-1A target genes. Moreover, skn-1a mutants or png-1 (a peptide N-glycanase required for SKN-1A function) exhibit a range of severe phenotypes when challenged by the aforementioned stresses.

The experiments are well executed and present a compelling case for involvement of SKN-1A in organismal adaptation to a loss of cytosolic proteostasis. In principle, the manuscript is an appropriate sequel to the prior work on proteasome dysfunction. There are several issues, primarily technical that we have raised below. Additionally, some of the claims made in the text need to be reevaluated.

1) The authors use two skn-1a specific null strains (mg570 and mg674 alleles) to show activation of proteasome subunits under genetic perturbation of the proteasome is skn-1a dependent. However, it is not clear why the authors chose to make a new mutant with the "an identical CRISPR lesion". Why was mg570 not sufficient? Have validation been performed to ensure this mutant behaves in the exact same way with the same phenotypes?

2) The authors use ts alleles of unc-54 and an a-beta model (bwm expressing) to conclude that skn-1a elicits a muscle specific upregulation of the proteasome. To make a stronger argument that this is indeed a direct result of skn-1a activation, authors could use skn-1a::GFP animals (GR2198) established in Lehrbach et al., 2016 to test whether skn-1a is stabilized in the nucleus in these proteasome activating paradigms.

3) The authors argue that the increased expression of rpt-3p::GFP reporter in the hsf-1(sy441) strain is evidence that "SKN-1A is broadly activated under conditions that increase the burden of unfolded proteins" (Figure S4). There is not enough evidence to support this claim as the authors fail to show skn-1a dependence of this phenotype. To provide additional strength for this claim, could the authors use RNAi (against hsf-1 in the skn-1a mutants or vice versa) to demonstrate skn-1a is required for this reporter phenotype? Additionally, 25C is usually lethal to worms of this genotype. If this experiment was done from larval stage at 25C, the result that rpt-3 expression is increased in sy441 worms may be a product of sampling only the most fit worms of this genotype and may not be indicative of the population of these worms.

4) The data in this paper provide a compelling narrative to support a novel role for skn-1a in maintaining cytoplasmic proteostasis in paradigms of protein misfolding. Unfortunately, a major weak point of their argument is the neglect to survey the role of the HSF-1, or other players of the canonical cytoplasmic heat shock response, in these paradigms. It is certainly possible that skn-1a does not work alone and actually relies on hsf-1 to promote longevity and improve phenotypes that result from misfolded protein expression. For example, the skn-1a mediated locomotion experiments (Figure 4) suggest skn-1a is important in maintaining proteostasis throughout age, especially when faced with protein folding stress. These experiments done with knockdown of hsf-1, however, would likely show an exacerbated phenotype and an increase in the age-related decline of locomotion. In order to declare these phenomena a distinct stress response, the authors should show that skn-1a mediated lifespan extension and skn-1a-dependent rpt-3p::GFP increases are independent of hsf-1 and/or reconcile the role the cytoplasmic HSR may be playing in these paradigms.

Reviewer #2:

This study investigates the molecular details of a proposed feedback mechanism for the transcriptional upregulation of proteasome subunits in *C. elegans*. A forward genetic screen was performed and several mutant alleles of proteasome components as well as unc-54 as inducers of a rpt-3p::gfp reporter were identified. Subsequently these authors showed this response to be cell-autonomous, activated by misfolded proteins, and dependent on the transcription factor Nrf1/SKN-1A.

This finding complements previous studies, which demonstrated that key components of the proteostasis network (such as HSP90) are increased in response to endogenous misfolded proteins. However, it remains unclear how non-native (misfolded) proteins lead to activation of endoplasmic reticulum-associated SKN-1A, and whether the amount of assembled proteasome complexes is actually increased (both levels and functional properties) in the presence of misfolded proteins. This study would also be strengthened by inclusion of biochemical experiments to support the proposed pathway.

Major comments:

1) The discovery of temperature-sensitive alleles in proteasome subunits in the mutagenesis screen is intriguing, and suggests that SKN-1A could indeed counteract age-related changes in proteasome activity. At what point does proteasome impairment occur in these mutant animals, and how this relate to the timing of rpt-3p::gfp induction? Proteasome function should also be directly assessed in worm extracts to measure proteolytic activity using fluorogenic peptides. It is known that ubiquitin-dependent substrates (such as UbG76V::GFP) accumulate only when the proteasome is almost fully blocked, therefore the dynamic range of this reporter needs to be carefully calibrated across different mutant backgrounds, perhaps by referencing to RNAi or a proteasome inhibitor.

2) It is puzzling that SKN-1A appears to be activated in the absence of impaired proteasome function, because this observation contradicts the proposed model whereby SKN-1A competes with misfolded proteins for proteasomal degradation. One would therefore expect to see reduced proteasome activity in the presence of misfolded proteins in skn-1a mutant worms, but this does not seem to be the case. This might be clarified by direct measures of proteolytic activity and in conditions in which UNC-54(ts) misfolds that can be monitored using biochemical approaches. Failure to detect differences in proteasome activity under extreme conditions (older age and higher temperatures) and in the skn-1A mutant would support the claim that proteasome function is not impaired in animals expressing misfolded proteins. Either way, it would be very important to know what happens to SKN-1A stability, subcellular localization, and binding to DNA in the presence of misfolded proteins.

3) It is unclear whether animal speed is a good proxy for protein misfolding as this is a very indirect measure; moreover, differences in movement can be linked to effects on egg-laying and many other types of complex behaviors. Instead, the authors might include other established measures of motility in the unc-54(ts) mutant strains, in addition to performing more detailed analyses of protein misfolding using biochemical and cell biological approaches. This is particularly important as some of the findings disagree with previous published results that established age-dependent misfolding of temperature-sensitive UNC-54 protein.

4) An important issue that remains unaddressed here is the mechanism by which misfolded proteins activate SKN-1A. Here, the temperature-sensitive alleles of unc-54 could be instrumental in determining the sequence of events that are required for the response, for example, by monitoring the levels and localization of SKN-1A upon protein misfolding. If SKN-1A protein becomes stabilized before proteasome activity is impaired, this could suggest that misfolded proteins compete for a yet unidentified component (perhaps a chaperone?) that is required for the continuous degradation of SKN-1A.

5) The existence of intra- and intercellular communication of stress responses is well-established in *C. elegans* and higher organisms, and may be a confounding factor in this study. Animals lacking SKN-1A show high levels of the UbG76V::GFP reporter in the intestine (Figure 3A), and therefore interrogation of a cell-autonomous pathway that appears to be specific to body-wall muscle in the skn-1a(mg570) mutant background is concerning. Along the same lines, analysis of the SKN-1A-dependent response in hsf-1(sy441) mutant animals (Figure S4) is also insufficient to support the notion of a cell-autonomous response to misfolded proteins in the cytoplasm.

6) There should be appropriate recognition that many seminal discoveries on the Nrf1/SKN-1A/CncC-mediated stress response were originally made in *Drosophila*.

Reviewer #3:

Lehrback and Ruvkun present interesting data suggesting that the ER-localized form of the transcription factor SKN-1 is in some way responsive to unfolded protein accumulation within the cytosol. In response to proteasome dysfunction or conditions consistent with unfolded/misfolded protein accumulation, the transcription factor mediates induction of proteasome genes. They present an interesting regulatory mechanism by which proteasome function (or maybe unfolded protein accumulation) relates to expression of proteasome component genes via the proteasome substrate SKN-1A

Overall, the manuscript is well written, the data is solid and the take home message is likely impactful. I have a few modest concerns, but in general am in support of the manuscript.

The data demonstrating that unc-54 mutants suffer a decline in movement on the first day of adulthood, but recover by days 3-7 are intriguing but somewhat preliminary. The authors have suggested this is due to improved proteostasis as the recovery requires SKN-1A. Is SKN-1A active during the phase of impaired movement? Is proteasome function reduced during this time? And, does it increase upon recovery? The authors have all of the reagents in hand, so the experiments ought to be straight-forward.

"Over-expression" of SKN-1A via the rpl-28 promoter is sufficient to prolong lifespan. It would be helpful to know how much over-expression is provided by the rpl-28 promoter.

Reviewer #4:

Skn-1/Nrf1 is a key regulator of the expression of genes encoding proteasome subunits. It is a highly regulated protein undergoing a remarkable itinerary. Previously it had been established that proteasome dysfunction increases the levels of active Skn-1, in a homeostatic feed-back loop promoting proteasome sufficiency. However, it remained unclear if proteasome insufficiency is the only mechanism activating Skn-1 or if other signals may prevail upon this transcription factor. Here the authors used rpl-28p::ub(G76V):GFP to measure proteasome reserve, showing that it is impaired in the positive control (myo3p::pbs-5[T65A]; encoding a dominant negative proteasome subunit), but not by the ts alleles of unc-54 or by the Aβ peptide and yet these insults activate Skn-1. These observations obtained through clever *C. elegans* genetics support the claim that skn-1 activation can proceed independently of proteasome dysfunction.

Given the significance of the claim and the convincing evidence in its favour, I support publication of this paper.

---

## [Author Response]

Essential revisions:Please consider the following 4 points in the revised version of the manuscript:1) Reviewer 2 points out that potential limitations to the dynamic range of the UbG76V::GFP reporter may compromise the strength of the conclusions drawn. The authors convincingly show that a mutation in the catalytic site of one of the catalytic proteasome subunits (pbs-5[T65A]) strongly stabilises the UbG76V::GFP reporter and that the cytosolic protein misfolding-inducing mutations that activate the Skn-1a target gene, rpt-3p::gfp do not share this feature (Figure 3). However, the concern remains that a significant deficit in proteasome capacity in the cytosolic protein misfolding-inducing mutations, might have been missed if the deficit fell under the threshold of the reporter's sensitivity. This concern arises because it is known that ubiquitin-dependent substrates (such as UbG76V::GFP) accumulate only when proteasome activity (as assessed by Suc-LLVY-AMC hydrolysis in cell lysates) is substantially blocked [see Figure 4a, from Dantuma et al., (2000))]. Therefore, it would be very reassuring to learn of a positive correlation between level of proteasome dysfunction, UbG76V::GFP reporter activity and Skn-1a target gene rpt-3p::gfp expression and then to learn that cytosolic protein misfolding is able to activate the Skn-1a target gene rpt-3p::gfp to similar levels without activation of UbG76V::GFP reporter activity. In other words that over a series of mutations/alleles, rpt-3p::gfp expression is correlated with UbG76V::GFP reporter activity when wrought by proteasome dysfunction but dissociated from UbG76V::GFP reporter activity, when wrought by cytosolic protein misfolding. RNAi, proteasome inhibitors or perhaps the numerous ts alleles in proteasome subunits isolated here (Figure 1—figure supplement 1) may be harnessed for this calibration (see point 1 of reviewer 2).

We used a dilution series of bortezomib to compare the sensitivity of the *rpt-3p::gfp* reporter of SKN-1A activation and the *Ub(G76V)::gfp* reporter of proteasome function. In these experiments we analyzed stabilization of Ub(G76V)::GFP in both wild type and *skn-1a* mutant animals, as compensation by SKN-1A may mask effects of low doses of the drug (just as our experiments examining the effect of Aβ and *unc-54ts* on Ub(G76V)::GFP included experiments in the *skn-1a* mutant background). We monitored expression of each reporter by fluorescence microscopy, using the same imaging conditions we had used previously. We examined animals raised on plates supplemented with bortezomib (i.e. chronically exposed throughout life). We think this provides the most meaningful comparison to our experiments with *unc-54ts* mutants and Aβ-expressing animals – which express the toxic/misfolding protein in muscle cells throughout development.

As expected, a high sub-lethal dose of bortezomib (40 ng/ml, ~100 nM) results in increased expression of *rpt-3p::gfp*, and in stabilization of Ub(G76V)::GFP. Bortezomib at this concentration is lethal to *skn-1a* mutants, precluding analysis of Ub(G76V)::GFP levels in the *skn-1a* mutant background.

Following treatment with 4 ng/ml bortezomib (~10 nM) *rpt-3p::gfp* levels are very slightly increased – this was the lowest bortezomib concentration that caused detectable *rpt-3p::gfp* activation. Following treatment with either 2 or 4 ng/ml bortezomib, levels of Ub(G76V)::GFP are unchanged in wild type animals. However, treatment of *skn-1a* mutant animals with 2 or 4 ng/ml bortezomib causes an obvious increase in Ub(G76V)::GFP levels. These data are included in Figure 3—figure supplement 1.

We draw two conclusions from these experiments: (1) compensation via SKN-1A-mediated upregulation of proteasome genes masks the effects of low bortezomib doses on proteasome function; (2) when SKN-1A-dependent compensation is removed, our UbG76V::GFP reporter is sensitive enough to detect the effects of very mild proteasome impairment that is below the threshold required to detect activation of SKN-1A using our *rpt-3p::gfp* reporter (i.e. 2 ng/ml bortezomib). Therefore, the results of this experiment support our suggestion that activation of SKN-1A by misfolded proteins may occur in the absence of impaired proteasome function. One caveat is that in these experiments we primarily monitor the response to proteasome impairment in the intestine, whereas the *unc-54* and Aβ experiments monitor the response in muscle. It is possible that the relationship between proteasome function, SKN-1A activation, and Ub(G76V)::GFP degradation may be different in different tissues. Thus, the activation of SKN-1A by mutant UNC-54 or Aβ may occur via weak impairment of proteasome function or another pathway of SKN-1A activation. We have changed our discussion to emphasize both possible interpretations of our results.

2) The evidence for Skn-1a activation by cytosolic misfolding presently rests on the Skn-1a genotype dependence of rpt-3p::gfp activity (Figure 1D,E). It would be helpful to buttress this convincing genetic argument with evidence that Skn-1a is activated at the protein level. Andy Dillin (reviewer 1) suggests this might be achieved by studying animals with skn-1a::GFP reporter (strain GR2198).

To test activation of SKN-1A at the protein level, we used a reporter strain ubiquitously expressing a version of SKN-1A lacking the DNA binding domain and fused to GFP at the C-terminus (*rpl-28p::SKN-1A[∆DBD]::GFP*). We previously showed that this form of SKN-1A undergoes the same post-translational regulation as full length SKN-1A (Lehrbach and Ruvkun, 2016). We used an extrachromosomal array containing the rpl-28p::SKN-1A[∆DBD]::GFP construct to simplify crosses between the reporter and the *unc-54ts* mutant strains.

We detected accumulation of SKN-1A[∆DBD]::GFP in body wall muscle cells in the *unc-54ts* mutants, but not in wild type animals. These data support our genetic evidence that SKN-1A mediates activation of the *rpt-3p::gfp* proteasome subunit reporter in the muscle of unc-54ts mutants. These data have been added to Figure 1—figure supplement 4.

3) The implicit claim that hsf-1∆-dependent activation of rpt-3p::GFP is Skn-1a-dependent (Figure S4) should be supported by evidence that such activation is blocked by a skn-1a deletion.

We have confirmed that activation of *rpt-3p::gfp* in *hsf-1* mutant animals is *skn-1a* dependent using an *hsf-1; skn-1a* double mutant. The new result has been added to Figure 1—figure supplement 3.

We also agreed with reviewer 1’s point that our measurement of *rpt-3p::gfp* expression in hsf-1 mutant L4 animals raised at 25°C may have been biased by their larval arrest phenotype. To remove any bias introduced by developmental defects, we raised animals of each genotype at 20°C. 20°C is permissive for development of *hsf-1* mutant animals. At 20oC, *rpt-3p::gfp* is not increased in *hsf-1* mutant L4s compared to wild type (Figure 4—figure supplement 2). We shifted L4 animals of each genotype (raised at 20°C) to 25°C for 24 hours before imaging and quantifying *rpt-3p::gfp* expression. This late larval temperature shift does not affect development of *hsf-1* mutants, so animals of all genotypes are fertile adults when imaged. Under these conditions we found that *hsf-1* animals show increased *rpt-3p::gfp* expression compared to the wild type. We also used these assay conditions to confirm the SKN-1A-depenence of this response. The acute activation of *rpt-3p::gfp* we observe under these conditions support our model that unfolded proteins activate the proteasome via SKN-1A.

4) The curious time course of the motility defect in the unc-54 mutant worms, suggests that correlating this phenotype with the temporal profile of Skn-1a activity might provide important support for the hypothesis that Skn-1a buffers the consequences of the misfolding-prone mutant. Therefore, we ask that you consider the specific suggestion of reviewer three and measure the time-dependent changes in Skn-1a activity in unc-54 mutant worms.

We compared expression of the *rpt-3p::gfp* reporter between day 1 and day 5 of adulthood in *unc-54ts* mutants. We find no change in expression of the reporter as a function of age in either the *unc-54(mg519*) or *unc-54(e1301*) mutant background. We have added these data to Figure 4—figure supplement 1. This result suggests that the strange time course of the *unc-54ts* mutants’ motility defect we have observed is not driven by age-dependent changes in SKN-1A activity.

Although our understanding of the mechanism of these effects is far from complete, our observations support the conclusion that regulation of gene expression by SKN-1A ameliorates cellular dysfunction caused by misfolded protein expression during aging. But we agree with the reviewers’ comments and have altered the Results section and Discussion section of the paper to present a clearer interpretation of these results.

Title:Please spell out "ER" [endoplasmic reticulum]

We have altered the title to spell out endoplasmic reticulum. We have also edited the title to include a reference to the longevity data in the paper. We want to make sure our paper attracts readers from the aging field in general as well as protein folding and homeostasis.